# A Survey on Open Radio Access Networks: Challenges, Research Directions, and Open Source Approaches

**DOI:** 10.3390/s24031038

**Published:** 2024-02-05

**Authors:** Wilfrid Azariah, Fransiscus Asisi Bimo, Chih-Wei Lin, Ray-Guang Cheng, Navid Nikaein, Rittwik Jana

**Affiliations:** 1Department of Electronic and Computer Engineering, National Taiwan University of Science and Technology, No. 43, Section 4, Keelung Rd, Da’an District, Taipei City 106335, Taiwan; superwilfrid@gmail.com (W.A.); d11002806@gapps.ntust.edu.tw (F.A.B.); mick.cwlin@gmail.com (C.-W.L.); 2Department of Communication Systems, EURECOM, Campus SophiaTech, 450 Route des Chappes, 06410 Biot, France; navid.nikaein@eurecom.fr; 3Google LLC, 1600 Amphitheatre Parkway, Mountain View, CA 94043, USA; rittwikj@google.com

**Keywords:** open RAN, O-RAN, open source

## Abstract

The open radio access network (RAN) aims to bring openness and intelligence to the traditional closed and proprietary RAN technology and offer flexibility, performance improvement, and cost-efficiency in the RAN’s deployment and operation. This paper provides a comprehensive survey of the Open RAN development. We briefly summarize the RAN evolution history and the state-of-the-art technologies applied to Open RAN. The Open RAN-related projects, activities, and standardization is then discussed. We then summarize the challenges and future research directions required to support the Open RAN. Finally, we discuss some solutions to tackle these issues from the open source perspective.

## 1. Introduction

The radio access network (RAN) is one of the main components in cellular networks [1]. The RAN connects the user equipment (UE) to the core network (CN). The RAN component has evolved throughout the years as a solution to the growing number of subscribers and rising user demands. The first version of RAN is distributed RAN (D-RAN), followed by centralized RAN (C-RAN), and lastly, virtualized RAN (vRAN). However, the vRAN solution hardly meets the expectations of the current 5G network requirements. The next generation of RAN is looking to open the interfaces in the RAN ecosystem.

The cellular network industry offers the next generation of RAN as a solution to fulfill 5G network requirements. Open RAN is proposed as an evolution of the vRAN [2]. To answer the challenges faced by vRAN, Open RAN breaks the closed nature of the previous RAN generations. Open RAN is essential for cost reduction in the RAN by promoting innovation that leads to competition among vendors due to its open nature. The architecture of Open RAN facilitates intelligent components within the RAN, driving innovation towards achieving energy savings and enabling dynamic power management [3]. Open RAN aims to deliver the expected quality of service (QoS) and quality of experience (QoE) of the latest 5G network requirement while preventing economic expenditure from skyrocketing. While advancing the previous RAN generations, Open RAN still faces challenges today.

The Open RAN movement and its goal to make an open-interface system have existed for the past few years. Even though Open RAN’s development has been around for years, there still needs to be more references or research that covers information about Open RAN holistically. The RAN advancement history and the Open RAN concept can be found in [1,2]. Open 5G network projects from the RAN and CN side, as well as their problems, are explained in [4]. The O-RAN Alliance architecture, summarized in [5], does not include the bigger picture of the Open RAN movement. O-RAN Alliance architecture is also discussed in [6,7], including its advantages and limitations that could be addressed in future research. However, these references could not serve the complete picture of Open RAN’s history, present condition, and what opportunities Open RAN can open up in the future.

This paper explains Open RAN technology in detail, emphasizing its further evolution from vRAN, as mentioned earlier. We first provide an overview of the larger Open RAN landscape, including the most current landscape of Open RAN in terms of projects and activities. Next, this paper describes the components of Open RAN and their implementation. The study focuses on O-RAN Alliance standards and architecture as a reference. This paper also explains the challenges faced in Open RAN and future research possibilities for next-generation technology. In addition, this paper will suggest some solutions to Open RAN challenges based on open-source approaches.

The rest of this survey paper is structured as follows: First, we explain why we need Open RAN in Section 2. Section 3 provides the background of the Open RAN movement and the technologies related to the Open RAN field. Section 4 reviews the projects, activities, and standards related to Open RAN. Section 5 discusses the O-RAN Alliance architecture. Section 6 describes the challenges and future research directions of the Open RAN field and the RAN technology in general. Finally, Section 7 addresses some of the challenges from an open-source perspective.

## 2. Evolution from Traditional RAN to vRAN

This section will summarize the evolution history of RAN. The real-world implementation of the RAN is better known as the Base Station (BS) [1]. Two major units of a BS are Radio Unit (RU) and Baseband Unit (BBU). RU is responsible for transmission and reception. Meanwhile, BBU is responsible for radio management, resource utilization, and other operations.

### 2.1. Traditional RAN or D-RAN

The first version of RAN was equipped with an integrated system between RU and BBU. BBU was usually installed in a room right below BS. RU could be installed in the room or at the top of a tower, enabling the RU to support connectivity in a large area [1]. In this context, the RU can also be called the remote RU (RRU). Either way, the distance between RU/RRU and BBU is short. This traditional RAN framework was called D-RAN [8]. Implementing D-RAN is straightforward as it does not require a high-speed interface between RU and BBU. Each RAN operates independently. The network densifies as the number of UEs increases and more BS are built.

Mobile Network Operators (MNOs) initiated a search for a solution to reduce Operating Expenditure (OpEx) due to substantial costs for renting the space for BS and the required cooling systems to operate the network, ultimately leading to the proposal of the C-RAN framework.

### 2.2. C-RAN

The “C” in C-RAN could be an abbreviation of “Centralized” or “Cloud”. However, both have the same idea: cluster the BBUs into a pool [2,9]. Initially, Centralized RAN is proposed to reduce the space rental cost and the power consumption of the air conditioner of the BBU by pooling BBUs from different BS into a single physical location. The fronthaul (FH) interface is introduced to connect the BBU pool and RRUs. Figure 1 shows how the FH interface (FHI) connects one BBU pool to many RRUs. Despite reducing the overall OpEx, Centralized RAN requires the FH to have high bandwidth and low time latency requirement [10,11].

The fundamental difference between Centralized RAN and Cloud RAN is the cloud system. In Centralized RAN, the BBUs are pooled in a physical location. In Cloud RAN, BBUs from each BS were pooled in a cloud server [1]. Cloud RAN is superior because the cloud control in Cloud RAN made it easier for the number of BBUs to be changed with time. The cloud also increased the baseband processing by exploiting general-purpose processors [2]. Cloud RAN further reduced energy consumption, increased network throughput, improved network scalability, reduced Capital Expenditure (CapEx), and reduced OpEx [8,12].

The C-RAN has been acknowledged as one of the most potential technologies to fulfill radio access’s 5G technical requirements [2]. However, it still has limitations, such as the huge FH overhead, trust problems, security problems, and single-point failure. These problems forced the C-RAN framework to shift its focus to advanced computing technologies. Some of these advanced technologies were virtualization and edge computing. Virtualization is the key to the shifting from C-RAN to vRAN.

### 2.3. vRAN

Virtualization means creating virtual instances over abstracted physical hardware [2]. In the context of vRAN, the virtual instances are network resources. Virtualization in RAN is heavily related to concepts such as software-defined networking (SDN) and network function virtualization (NFV) [8]. Simply, vRAN brings virtualization to Cloud RAN. In the cloud server, multiple virtual BBUs (vBBUs) are deployed. vBBU can be deployed on a Virtual Machine (VM) or container. This system enables orchestration and resource scaling in vRAN. Ease to scale up and down network resources led to lower energy consumption, dynamic capacity scaling, efficient use of network resources, improved service reliability, and better service quality [2,13]. Approximately 50% of data-processing resources required can be reduced when virtualization is applied on Cloud RAN [14].

Overall, vRAN minimizes the operational and investment costs for MNOs. However, vRAN has to deal with the properties of the wireless channel. Network resources must be shared and distributed fairly and efficiently to different RRUs while considering QoS requirements. This made the vRAN system to be significantly complex. vRAN also still uses proprietary interfaces that prevent interoperability and a multivendor environment. As a result, vendor lock-in and monopoly prevent the network equipment price from being cheaper.

## 3. Open RAN Movement and the Related Technologies

This section will introduce the history of the Open RAN movement and the technologies used to build Open RAN. The Open RAN term had been proposed since May 2002 [15]. However, the current widely known Open RAN term is an industry movement in wireless telecommunications to develop disaggregated and interoperable RAN [16]. Disaggregated and interoperable are the two fundamental pillars of Open RAN. The two fundamental pillars will significantly benefit us because Open RAN’s architecture will enable a multivendor environment and can run on general-purpose processors. In the next subsections, we will explain the technologies that support Open RAN in detail.

### 3.1. Open RAN Movement

The growing cellular network market demand forces MNOs to optimize each network component, including the RAN. Several challenges should be addressed to deploy a good and reliable RAN. One of them is how to meet specific QoS requirements. 5G introduces new three use cases domain, namely Enhanced Mobile Broadband (eMBB), Ultra-Reliable Low-Latency Communication (URLLC), and Massive Machine Type Communication (mMTC) [17]. Every application and every wireless device has its own specific QoS requirements [1]. The existing monolithic RAN could not satisfy the diverse requirements of these use cases. Centralizing all use cases on a single network due to the limitations of the monolithic RAN would reduce both QoS and QoE [18].

Network upgrade for enhanced flexibility and adaptability is essential to meet different QoS requirements for each application. This flexibility can only be achieved by disaggregating RAN components [1]. It enables the RAN to behave differently to each application’s particular QoS requirement, leading to a more intelligent and versatile RAN. RAN’s architecture should support multiple levels of QoS in handoff scenarios at one time [15].

Another challenge for RAN is combining the existing NFV with artificial intelligence (AI). The trend of NFV has existed since the development from C-RAN to vRAN. However, the virtualization concept proposed in vRAN is complicated, needing mechanisms for management and orchestration [2]. Management and orchestration can be performed by leveraging AI. RAN needs NFV with AI embedded in it. As we know, AI has impacted the computing and networking world significantly until today [19]. AI can be used to analyze the enormous amount of RAN-generated data. Then, the information gathered can be used to anticipate problems in the network and take necessary action to ensure network QoS is delivered to the users. AI can make the NFV system more intelligent, optimized, and improved. Further explanation about NFV can be seen in the next subsection.

It has already been stated that RAN’s improvement is happening by introducing cloud-native principles. We shall develop the cloud-native approach by improving RAN’s software and hardware. The previous generations of RAN had a similar weakness: MNOs needed more control over their software and hardware innovation. Before Open RAN, RAN’s software and hardware were proprietary and remained optimized to one vendor only [16]. It caused an economic monopoly with extreme overhead and led to unsupervised systems because the only vendor that could supply the hardware and software was that one vendor. MNOs depend heavily on vendors to provide innovation and new features on RAN technologies. Let us take AI and NFV in the previous paragraph as an example. AI-embedded software is needed to complete the NFV advancement process. However, MNOs could not implement AI-embedded NFV because they depend on whether their partner vendors offer this solution. So, before MNOs can freely innovate their services, they must regain control of their network components.

The remaining challenge for RAN involves the need for efficient interconnection among components within the network architecture [18]. It was caused by the uneven standards used for cellular networks around the globe due to the proprietary interfaces used. Because of these different characteristics of each vendor’s interfaces, hardware and software from different vendors could not interconnect easily. This led to plenty of waste of wireless infrastructures and spectrum resources. When MNOs try to ignore these differences and treat all the cellular networks the same, their users will experience low QoS and QoE of the network.

Another fundamental pillar of Open RAN is disaggregation. Generally, disaggregation means the software is detached from its hardware [16].

Vendor develops their software to make it compatible with hardware from different providers. Through disaggregation, RAN software can run in general-purpose processors in COTS hardware, further extending its interoperability. RAN’s software can now run on any vendor’s hardware. Because softwarization is performed in RAN, acceleration techniques in software development can also be implemented, such as continuous integration and continuous deployment (CI/CD). In turn, RAN’s time to market and deployment time can be reduced. Disaggregation and softwarization will be explained in further detail in the next subsection.

Open RAN has open interfaces in its architecture, which has become a fundamental pillar.

An open interface means these connections are standardized, and everyone can see the specifications freely. One of the most notable examples of an open interface is the internet, where everyone uses the same protocol to communicate. Other terms have the same meaning as the “open interface” terms, such as “open API”, “open standard”, “open technology”, or even just “open”. When a vendor declares that their solution is “open”, it usually means that the vendor offers an open interface.

Open source software is characterized by its accessible source code, written in programming languages, allowing programmers to create or modify their software in a non-proprietary and collaborative way. It has an Open Source Initiative compatible license that allows anyone with the software’s source codes to re-distribute, modify the software, or even derive codes from the software [20]. In the RAN context, open-source code RAN software means that the software source code to run the RAN network function can be accessed by everyone. Open-source RAN software is available. However, it is a relatively immature and comparatively limited feature set compared to commercial alternatives.

Rakuten has proven to be the world’s first, which is currently the only commercial-scale nationwide network that uses Open RAN standards and architectures [21]. It was supported by Umlaut, who evaluated several performance metrics by comparing Rakuten Mobile data in Tokyo to other MNOs in major cities worldwide. The report is decidedly positive, scoring 920 out of 1000 and a “very good” rating. Compared to other cities, Rakuten can provide connectivity at a much higher speed than other MNOs. It proves that applying Open RAN in network architecture improves network connectivity performance.

Besides improving performance, Open RAN can also make the network connectivity be more cost-efficient. Because the framework includes AI in its architecture, the Open RAN framework will be able to facilitate automated operational network functions [2]. Open RAN is built to perform some complex tasks that had never been done before by its predecessor. The existence of machine learning (ML) and AI reduces RAN dependency on human force, thereby reducing operational costs. Cost reduction will be much more significant in the future with the development multivendor environment. The disappearance of a monopoly relationship between hardware and software in the Open RAN framework is designed to enhance the cost efficiency. MNOs can change a particular part that needs an upgrade or is broken without needing to change the whole network [16].

Rakuten serves as the proof. Rakuten Mobile 4G tech’s CapEx and OpEx are approximately 40% and 30% lower than traditional RAN deployment’s expenditures [21]. Furthermore, it is also mentioned that the cost spent by Rakuten is much more efficient when deploying a 5G network, saving up to 50%. This cost efficiency can benefit rural regions to enjoy economical network connectivity. These two advantages can be reached through Open RAN’s openness and intelligence fundamental pillars. By solving the previous RAN problems, the network’s performance will be improved and its cost will be more efficient.

Open RAN started to develop rapidly in these last six years. The first development of Open RAN occurred in February 2016, when Telecom Infra Project (TIP) formed the OpenRAN Project Group [16]. Since then, TIP has brought together more than 500 MNOs, suppliers, vendors, developers, and integrators that are using open source technologies and open approaches [16,22]. Further details about TIP OpenRAN project group are written in the next section. A year after the formation of the community, the first trials of Open RAN started in India and Latin America [16].

### 3.2. Technologies Related to Open RAN

To make an open and intelligent RAN architecture, Open RAN is supported by several technologies or approaches. Disaggregation, SDN, NFV, functional split, cloudification, automation, intelligence, network slicing, open source, and mobile edge computing (MEC) are some of them. These technologies and approaches will be introduced briefly in this subsection.

RAN disaggregation can refer to the separation of RAN into software and hardware or dividing it vertically and horizontally [23,24]. The former is a separation process between the network’s software and hardware that occurred in a network architecture. As explained before, RAN’s software and hardware used to be proprietary and integrated. Through disaggregation, the term RAN’s functionality softwarization emerges where the software does not depend on specific hardware and vice versa. The main differences between non-disaggregated and disaggregated network architecture are shown in Figure 2. Software used to be integrated with hardware, whether it is in the CN or RAN. In a disaggregated network architecture, software and hardware are now separated. Through softwarization, the further separation of control and data is carried out, thus the the term SDN emerges [4,25,26].

Vertical and horizontal RAN disaggregation is also known as functional splits. Horizontal functional split is a selection process of the appropriate centralization level in RAN framework [27]. It separates the integrated BBU into two separate units: Central Unit (CU) and Distributed Unit (DU). Horizontal functional split also refers to the DU and RU separation. The degree of centralization in the horizontal functional split is flexible [28]. However, trade-offs should be considered when choosing these functional split options [11]. On the other hand, the vertical split is the separation between the CP and UP of the RAN. The CP and UP splitting (CUPS) is the extension of the SDN concept. The difference between vertical and horizontal split can be seen in Figure 3. Further explanation about functional splits will be discussed in Section 4.

Virtualization enables the softwarized RAN to be sub-divided into smaller parts within single hardware that enhances softwarization [29]. There are two virtualization technologies, which are hypervisor-based and container-based virtualization [2].

Virtualization allows hardware to host functions of multiple virtualized units, leading to a term called NFV, which virtualizes all network services, such as virtual CU or virtual DU. These virtualized functionalities are called Virtual Network Functions (VNFs) and they run on top of VMs. NFV uses hypervisors named Virtual Machine Monitor (VMM) or virtualizer. The VMM hosts and runs Virtual Machines (VMs) that host VNFs.

Another related technology is called network slicing. Network slicing branches from NFV. Network slicing is the concept to slice or partition the physical network to form virtual resources [4,30]. Slicing is performed in all parts of the network. Through the network slicing, there are multiple virtualized network that lies in the same physical network. Each virtualized network can flexibly be allocated for different use cases requirement in 5G. Different services can be allocated to each of the slices to satisfy specific user needs. Network resources can also be allocated dynamically according to each slice’s needs. Network slicing contradicts the former “one size fits all” service model.

Disaggregated RAN allows its software to operate on different hardware. MNOs can use commercial off-the-shelf (COTS) server or a cloud instead of using dedicated hardware to run their software. This condition is called cloudification, where the hardware’s roles are replaced by the cloud or COTS. MNOs might choose to use cloud or COTS because the deployment cost is cheaper than investing in their own hardware to run their software. There are also several types of cloud deployment. Choosing the right cloud deployment is the first step a MNO should make in cloudification. MNOs might want to choose public cloud deployment, where the cloud resources are owned and operated by a cloud service provider. A public cloud may have lower costs and be easier to manage, but the MNO has less control over the cloud resources and the level of security might be low. MNOs can also choose private cloud deployment, where the cloud resources can only be used exclusively by one business or organization. Private cloud deployment allows for more flexibility, customization, and control. However, it requires advanced technical skills and is more costly. MNOs can also choose the hybrid cloud deployment, where data and functions move back and forth between private and public environments.

Since there is a cloud deployment option, MNOs can choose different approaches to provide their service, namely a brownfield or a greenfield strategy. In the greenfield strategy, they have to make their network from the ground, which means that the MNOs should deploy the software and hardware from scratch. They do not use existing third-party cloud infrastructure. After they do the groundwork, they move those network components to the new cloud infrastructure. The greenfield strategy is the opposite of the brownfield strategy, where many of a network’s functions of the previous architecture are retained. Simply, it means that in a brownfield project, the MNO needs to upgrade or add new features to an existing cloud network and use some legacy cloud components. Brownfield projects are mostly carried out after a greenfield project, with the purpose of developing or improving an existing application infrastructure. MNOs should consider trade-offs between the two approaches.

Through softwarization, the RAN’s characteristics and behaviors are easier to reprogram, thus making the RAN programmable. The advantages of Open RAN programmable behavior is its network performance will be more optimized, the network resources can be dynamically allocated, the software and hardware functions can be controlled automatically, and novel algorithm can be leveraged to improve their own networking system performance [31]. The more advanced technology of programmability is called automation, where characteristics and behaviors of RAN are all reprogrammed automatically not by humans, but by computer programs.

There are two types of automation, namely orchestration and management. Orchestration is the automation of the softwarized RAN deployment process. Management automates RAN monitoring, configuring, coordinating, and task managing. The management system of RAN is called radio resource management (RRM). From RAN management, there is a new term called self-organizing network (SON). With SON, the RAN is requested to do its self-configuration, which includes new deployment of nodes, performance optimization, and fault management [32]. The intelligence of RAN is proved through AI/ML embedded in the RAN. AI/ML turns RRM into a more intelligent one, namely RAN Intelligent Controller (RIC) [2]. The RIC facilitates the optimization of RAN through closed-control loops between RAN components and their controllers [33]. RICs have made the Open RAN more adaptable, effective, and economical. As shown in Figure 4, these loops operate at different times, ranging from 1 ms to thousands of milliseconds. Real-time RIC (RT RIC) loop works for less than ten milliseconds. Near-RT RIC loop time range is between 10 ms to 1 s. The Near-RT RIC is responsible for deploying AI-enabled optimization and giving feedback for UEs and cells. The Non-RT RIC loop time typically spans 1 s or more, and it is responsible for releasing AI-enabled service policies and running analytics for the entire RAN.

Even though not mandatory like open interface, open source accelerates improvement of network infrastructure. In the past, RAN software has been largely proprietary and developed for specific hardware. The virtualization concept introduced by vRAN caused a significant change in the way network systems were designed. This change has significantly increased the use of server-based platforms and virtualized network functions in vRAN, and also in Open RAN, thus opening new chances for companies and network vendors to rebuild their RAN hardware and software to server-based platforms based on open-source software. Simply, open-source software has become a major solution to overcome challenges for RAN’s software infrastructures. Its big contribution to RAN’s software caused O-RAN Alliance and other communities and groups to develop open source solutions and adopted them to Open RAN’s significant functions. Although these open source solutions are still in the early deployments stage, the telecommunication industry is mostly expecting companies to use open source-based software infrastructure solutions to run RAN applications.

Another related RAN technology is the smart network interface card (NIC). A smart NIC is a programmable accelerator that can increase the efficiency and flexibility of the data center network, data security, and data storage. Smart NIC can offload computation from its host processor [34]. Smart NIC can transparently unpack virtual switching data path processing for networking functions, such as network overlays, network security, load balancing, and telemetry. Through the use of Smart NIC or accelerator cards, a regular server performance and latency can be enhanced to meet the requirements needed to function as vBBU. Xilinx T1, T2, and T3 Telco Accelerator Card are some example of accelerators.

The other technology related to Open RAN is MEC. MEC is a cloud service that runs at the edge of a network. It performs several specific tasks that would be processed in centralized core or cloud infrastructures. MEC moves the computing process closer to the users for enabling applications and services requiring some specific network characteristics that differ from other applications or services. MEC is capable to move content and functionalities to the edge; providing any private cellular network services by using its localized data process; deploying computational offloading to IoT devices; leveraging the proximity of edge devices to end users; and enhancing the privacy and security of mobile applications [4,35]. Until today, there are several researches remaining for MEC, some of them are binary and partial offloading for MEC; MEC resource management system; MEC-open RAN-network slicing integration and their combined orchestrators; and MEC-embedded-RIC or called inter-near-RT RIC [32,35].

Until this part, we already knew that Open RAN is related to many technologies. These technologies related to Open RAN can improve costs in the future 5G network [36]. The term used to measure the total cost spent for deploying a network is called total cost of ownership (TCO). TCO is the sum of CapEx and OpEx needed when deploying a network. The reports also mentioned that the average TCO reduced from implementing these related technologies is 20% if we compare it to 4G’s TCO. The potential savings can surpass 20%, as a 5G network integrating these advanced technologies can generate significant business benefits essential for any enterprise.

SDN and NFV: The deployment of SDN and NFV concept in 5G network can save for about 25% 5G’s TCO. This 25% savings are covered both from RAN and core virtualization. In the future 5G era, NFV will be a potential-generating tool, and maybe people will heavily rely on NFV when using 5G network.Automation and intelligence: Automation and intelligence are heavily related to AI. AI in 5G will be expected to save about 25% 5G’s TCO. We already know that AI in 5G has made Open RAN able to do many things, including automation and deploying ML. However, AI applications in cellular telecoms for now, are relatively rare. While this is expected to evolve over time, operators should acknowledge this situation when strategizing for the near future.Network slicing: Network slicing will not be a part of a cost-saving element in 5G, but the one that can reduce cost is network sharing. Network sharing can make savings for about 40% TCO.Cloudification and open source: The deployment of cloud and open-sourced software in 5G network can save up to 5% 5G’s TCO.

The cost efficiency mentioned in this section can also be further improved by implementing Open RAN in 5G networks. Before Open RAN era, the previous RAN hardware infrastructure was the part that had the highest cost of all parts in RAN infrastructure. Basically, to make good RAN hardware, there must be cabinets, radio antennas, baseband processing tools, power tools, cooling tools, and other tools. These tools had made for about 45% until 50% out of RAN’s TCO. In 5G technology, the infrastructure cost has increased by 65% in some deployments. This 65% cost is probably the maximum possibility for any Open RAN scenario. This cost can be less than 65% for some Open RAN’s lower-cost deployment scenarios.

This wide gap depends on some modifications used in the 5G Open RAN architecture. One of those modifications is adopting C-RAN’s architecture deployment into the Open RAN. At least, this C-RAN adoption has saved 25%, compared to the previous generation of C-RAN cost, which is D-RAN. While C-RAN adoption has saved 25%, this number is predicted to be up to 45% in the future.

While the FH has made its way to reduce the RAN’s TCO, the backhaul will not be similar to FH. Because 5G connectivity requires higher capacity with lower latency, backhaul has its cost increased by 55% in some higher Open RAN deployments. This 55% number is not definitive, because the increased cost can probably be lower than this number in some lower Open RAN deployments.

## 4. Projects, Activities, and Standards Related to Open RAN

This section describes some important Open RAN related projects and activities. The term “related projects” means how Open RAN is implemented in several contributions and other projects from organizations and companies who believe in the vision of Open RAN. However, even if these projects implement the Open RAN framework, they may have different standardization and do not necessarily follow the O-RAN Alliance Standard. We will also explain some mobile communication standards that are tightly related to open RAN.

### 4.1. Projects and Activities

The first project that we will explain is the O-RAN Alliance. This community name is shortened to O-RAN. Because the term is tightly related is Open RAN, many people assumed that O-RAN, like ORAN, is also a shorter version of Open RAN. In addition, those two terms are used by industries interchangeably. There are also many journals that used these two terms at the same time, and with the same meaning. That is not the case. To make it clear, O-RAN is the short form of O-RAN Alliance, a name of a software community [16].

O-RAN Alliance is a community that has tried to standardize and detail the Open RAN specifications so they can be implemented in real life [37]. O-RAN Alliance was founded in February 2018 [38]. To form O-RAN Alliance, five big MNOs at that time gathered, which were AT&T, China Mobile, Deutsche Telekom, Orange, and NTT Docomo. The main purpose of making O-RAN Alliance is to promote an open and intelligent RAN. This alliance is a combined version of xRAN Forum and C-RAN Alliance [16,38]. All these communities were involved in deploying open interfaces, big data intelligent control, and virtualization in RAN. These communities had also recognized the need to make an open network. After these communities were merged into O-RAN Alliance, these communities had become official members of O-RAN Alliance, but they still kept their original purposes to deploy a more reliable and faster network. The first O-RAN Alliance board meeting was held in June 2018, and the first work group (WG) meeting in September 2019 [38].

The O-RAN Alliance has its own core principles. Basically, there are two core principles of O-RAN Alliance. These core principles are openness and intelligence [2,39]. To achieve these two high-level core principles, O-RAN Alliance has given some reference designs about how the architecture of the Open RAN should be [38]. These reference designs from the O-RAN Alliance are called O-RAN vision, which consists of standardization design, virtualization design, and white box design [39,40]. To be able to achieve the three visions, the O-RAN Alliance divides itself into smaller groups. There are two kinds of groups in the O-RAN Alliance. The first groups are the groups divided based on each work description, called WGs. There are ten WGs, and these ten WGs’ objectives and chairpersons can be seen in Figure 5. The second groups are groups divided, not based on each work description, but based on their scopes or their focuses. We call these second groups focus groups (FGs). There are six FGs [37]:Standard Development Focus Group (SDFG): SDFG is responsible to make the standardization of O-RAN Alliance and to make the main interface to other Standard Development Organizations (SDOs) that will be relevant for the alliance’s work. SDFG also coordinates incoming and outgoing liaison statements.Test and Integration Focus Group (TIFG): TIFG’s task is to define O-RAN Alliance’s overall approach for doing tests and integration, including coordination and specifications tests for WGs.Open Source Focus Group (OSFG): OSFG is responsible for dealing with open source-related issues for O-RAN Alliance. OSFG is the parents of OSC, which will be explained briefly.Industry Engagement Focus Group (IEFG): IEFG is responsible for promoting, accelerating the adoption and innovation of O-RAN-based technology in industry and O-RAN Ecosystem engagement.next Generation Research Group (nGRG): nGRG is responsible for researching open and intelligent RAN principles in the 6G system and beyond.Sustainability Focus Group (SuFG): SuFG is responsible for creating energy-efficient and environmentally friendly mobile networks.

To maintain the healthy relationships among WGs and FGs, the O-RAN Alliance formed a special committee, whose position is above all of these WGs and FGs. This committee is called the O-RAN Technical Steering Committee (TSC). TSC consists of member representatives and the technical leader from every WG, and this community represents both the members’ side and the contributors’ side in making decisions for every technical innovation made by WGs and FGs for the O-RAN Alliance. TSC is tasked to provide technical guidelines to every WG and FG, and to approve O-RAN’s specifications, which were created by WGs and FGs based on members’ approvals and publications [37].

Another important committee within the O-RAN Alliance is the MVP Committee (MVP-C). MVP-C is relatively new in O-RAN Alliance. MVP-C’s main job is to prepare a MVP for O-RAN’s WGs and the public. MVP will provide a priority list of work items and is used to coordinate WGs. MVP helps the O-RAN Alliance to work more effectively. For the public, MVP gives a clear understanding of O-RAN’s roadmap and priorities.

As mentioned earlier, the third FG, called OSFG, serves as the parent entity for OSC. OSC was founded in April 2019 as a collaboration between O-RAN Alliance and LF [16,41]. This collaboration aims to support the creation of RAN’s specific software. OSC is responsible for dealing with open source-related issues for O-RAN Alliance [37]. The software community is focused on aligning a software reference implementation with the O-RAN Alliance’s Open RAN architecture and specifications [42]. Due to this focus, OSC’s responsibilities are developing the open-source software, coordinating with other open-source communities, promoting the open-source software, and addressing wireless technology support for essential patents [37,41,42].

The workflow of OSC can also be seen in Figure 6. In the figure, we can see that OSC and the O-RAN Alliance work together in a loop. Firstly, 11 WGs contribute to making RAN specifications, architectures, and reference designs. These specs and architectures are checked to 3GPP and 5G network standards. Besides checking to 3GPP and 5G standard, these specs and architectures are also given to TIFG. TIFG tests whether OSC’s software aligns with the specifications defined by the O-RAN Alliance. If there is any OSC code that works differently than the original specification, TIFG will request TOC to resolve these variance problems so that it aligns with the standard specification. After that, TOC prioritizes and negotiates with the O-RAN Alliance about solving these variance problems. The result of this negotiation is passed to the Requirements and Software Architecture Committee (RSAC). On the other hand, the O-RAN Alliance’s WG1 also gives recommendations to RSAC about RAN specs or design features for inclusion in a particular release. Until this process, we can see that RSAC receives two things: negotiations from TOC and specs recommendation from WG1. When receiving the inputs, RSAC selects those recommendations based on available resources and software release timeline. Therefore, RSAC and OSC together develop their release planning. This release planning is implemented by OSC to produce the software releases every 6 months. OSC also interacts with other open source communities, as illustrated in the figure, such as Open Network Automation Platform (ONAP), Akraino, and Acumos AI. OSC contributes to these communities and at the same time, OSC also uses the software made by these communities. From the software release, the process starts again.

The O-RAN Alliance supplies further support to the Open RAN movement by providing OTICs and conducting the O-RAN Global PlugFest. OTIC provides a working laboratory environment for Open RAN E2E system testing, certification, and badging [43,44]. OTIC’s environment is designed to be vendor independent, open, collaborative, and secure. Currently, there are 15 OTIC labs around the globe [43] that usually held PlugFest to do interoperability test among vendors. PlugFest is a worldwide testing and integration event to demonstrate the O-RAN ecosystem [43]. The scope of PlugFest includes testing using the O-RAN Test Specifications; Validation and Demo of the O-RAN architecture elements, concepts, feature packages, reference implementation, and reference design; and O-RAN certification and badging dry-run.

Besides O-RAN Alliance related projects and activities, there are other projects that stem from the Open RAN movement. OpenAirInterface (OAI) is one of them. Started in 2009, OAI is an open software for RAN and CN developed by Eurecom. Since 2014, OAI is managed by OAI Software Alliance. This alliance originated in France and is divided into several project groups. OAI 5G RAN project group aims to develop an open source 3GPP compatible 5G next generation node B (gNB) RAN stack software [45] for its community members. Currently, OAI provides software-based implementations of evolved node Bs (eNBs), UEs and Evolved Packet Core (EPC) that are suitable with Long-Term Evolution (LTE) Release 8.6 [4]. Besides OAI 5G RAN, OAI also has the Mosaic5G project group with other projects, such as FlexRIC, FlexCN, and Trirematics.

OAI RAN’s source code is written in C programming language, and this source code is distributed under OAI Public License, which is the combination of Apache License v2.0 and fair, reasonable, and non-discriminatory (FRAND) clause [4,46]. In this OAI framework implementations are compatible with Intel x86 architectures running the Ubuntu Linux OS. These implementations also require some modifications, mostly in kernel and BIOS-level modifications. These requirements should be fulfilled to make the OAI-RAN platform can perform in real-time, including installing a low-latency kernel, disabling power management, central processing unit (CPU) frequency scaling functionalities, and can also be used for field experimentation and evaluations with emulated wireless links [4,47].

The second related project is srsRAN. srsRAN is a free open-source software radio suite for 4G and 5G started in 2014 [48]. The project was formerly known as srsLTE and was originally developed by a startup called Software Radio System. Originally, srsLTE provided software implementations of LTE. Similar to OAI-RAN, the software implementations performed by srsLTE is in eNB, UEs, and EPC are suitable for LTE Release 10 [4]. The additional features for srsLTE are worked in 3GPP’s Release 15. Also similar with OAI-RAN, srsLTE’s software codes are written in C and C++ programming languages. srsRAN codes are distributed under GNU Affero General Public License (AGPL) version 3 or commercial license. The software is also compatible with Ubuntu plus Fedora Linux distributions. Different than OAI-RAN, srsLTE does not require kernel or BIOS-level modifications. srsLTE does need to disable the CPU framework scaling to do the real-time performance. srsLTE has now transformed to srsRAN as the project expands focus to 5G New Radio (NR) [48].

The third project is the TIP OpenRAN Project Group. The project group was started in February 2016 [16]. At that time, vRAN was widely discussed but still, that version of vRAN was considered impractical to implement and deploy for commercial traffic [31]. The project group consisted of MNOs who felt that the telecommunication industry was lacking innovation and the equipment needed for deploying network connectivity were extremely costly, especially in a highly concentrated or closed ecosystem [16]. TIP OpenRAN project group is an initiative to develop RAN solutions based on an open interface that can run on general-purpose hardware [31]. This project’s scope includes multiple generations of the mobile communication system, from 2G to 5G. However, unlike previous projects, this project is closed source [4]. OpenRAN project group has goals to encourage innovation through building an ecosystem that can enable openness of the network; enable multi-vendor and software-based RAN; and reduce network deployment and maintenance costs [16,31]. All work carried out by TIP OpenRAN Project Group is reviewed and approved by TIP technical committee. The members of the project group are mostly companies who have become members of TIP. They include Vodafone and Telefonica, along with Intel as TIP co-chair [31].

The fourth project is ONAP. ONAP is an open source project started in 2017 that provides a common platform for telecommunication companies, providers, and MNOs to design, implement, and manage differentiated services [49]. ONAP project is one of LF projects [4] which is linked to Open RAN, primarily through the SMO project in OSC [50]. The project automates 5G by using SDN and NFV technologies. ONAP includes all the Management and Orchestration (MANO) layer functionality that compliant with NFV architecture from European Telecommunication Standards Institute (ETSI). ONAP includes network service design framework and fault, configuration, accounting, performance, and security (FCAPS) functionality.

The fifth Open RAN related project is SD-RAN. SD-RAN is one of ONF’s projects. ONF is a community of MNOs that contribute to their own open source codes and frameworks for their networks’ deployments [4]. ONF always works closely with O-RAN Alliance, TIP, Broadband Forum, LF, and Open Compute Project in every network deployments [24,51]. Started in 2020, SD-RAN’s purpose is to build open source RAN components. In particular, SD-RAN is an ONF’s component project that focuses on building the near Real-Time (RT) RAN Intelligent Controller (RIC) and xApps [4,6,52]. The xApps-SDK code made by SD-RAN is shared with OSC. SD-RAN has made its initial release in January 2021, called SD-RAN v1.0.

In addition to the projects above, there are also testbeds that can instantiate software 5G networks by leveraging some of the open-source components above. POWDER-RENEW, COSMOS, and Colosseum are a few examples [4]. The three testbeds are part of the Platforms for Advanced Wireless Research (PAWR) program [4,53]. POWDER-RENEW and COSMOS have worked with O-RAN and can support indoor and city-scale outdoor scenarios [4]. On the other hand, Colosseum is advertised as the world’s most powerful wireless network emulator. Colosseum can support a large-scale network emulator scenario with 256 programmable software radios [4,53].

### 4.2. 5G, 3GPP, and Open RAN

3GPP is a standard development organization for the cellular telecommunications technologies [54]. It was assigned to work with the International Telecommunication Union Radiocommunication Sector (ITU-R) on scheduling and formulating the 5G technology standard, as ITU-R is the authority for radio communication [55].

The 3GPP releases introduce the new three use cases domain of 5G, which are eMBB, URLLC, and mMTC [56]. From these releases, 3GPP introduced a new term called NR. NR is a new radio access technology (RAT) that 3GPP developed for 5G technology. Simply, NR is the term for 5G RAN. 3GPP defines some specifications for the 5G NR technology, such as Standalone (SA) configuration, Non-Standalone (NSA) configuration, CU-DU functional split, and CUPS [55,57].

The 5G NR configuration can support either SA or NSA configuration. As the name implies, the NR SA means the 5G NR framework works independently without having another generation technology connected. NR NSA is the opposite of the NR SA. The NR NSA still connects to 4G CN or EPC to support the NR. 3GPP also defines the interface differences between NSA and SA configurations. SA and NSA have their CN and RAN involved differently in each configuration:NSA Configuration: The NR’s eNBs connect to the gNB through an X2 interface. Another interface used in NSA is S1 interface, connecting eNBs and gNBs to EPC. NSA uses EPC as its CN, and its users can connect both to eNB or gNB to deliver the network.SA Configuration: SA enables operation service to be provided solely on the basis of gNB. The gNB connects to other gNBs with Xn interface and connects to the 5G core (5GC) through NG interface. SA uses 5GC as its CN. SA’s UEs will only connect to gNB. CU and DU are parts of gNB.

There are two kinds of functional splits: horizontal and vertical split. The CU–DU functional split is called horizontal functional split. 3GPP defines the interface between the CU and DU as the F1 interface. Another further specification that 3GPP defines is the vertical functional split. A vertical functional split is performed to separate the UP and CP radio protocol. UP carries the network user traffic. UP protocol stacks consist of three layers, which are packet data convergence protocol (PDPC), radio link control (RLC), and medium access control (MAC). CP radio protocol contains radio resource control (RRC) layer which is responsible for controlling or configuring the lower layers, broadcasting information from a terminal to a cell, and deploying RRC connection functions. Further details about UP and CP can be seen in [57].

After many years of planning, the 5G technology became a reality in 2019 [36]. The 5G launch already existed for about 36 deployments across Asia, Europe, and North America. The telecommunication companies had also worked hard to implement 5G in real life. However, the 5G technology still has so many elements that need improvements today. One of these improvements should be optimizing 5G network costs. As we have seen from the previous section, the Open RAN movement was also started due to this concern.

Previously, we have explained how 3GPP defined the RAN and its interfaces standard. We describe how 3GPP defined RAN and interfaces where they are open and standardized, such as the Uu, S1, and X2 interfaces [16]. Although 3GPP has provided the standardization for 5G RAN, it is still insufficient to provide a clear and definite standard for open interfaces. For example, no standardization defined for FHI. Another example is that some interfaces are defined as optional, therefore implementation varies between vendors.

The unclear RAN standard was not enough to provide interoperability for existing mobile terminals and smartphones, especially if a multivendor environment was introduced. It was necessary to create a new movement in the telecommunication industry by cooperating with other companies in industries and releasing new RAN functions and interface standards that can enhance its value. This is why the open framework network movement was started. To fulfill this necessity, NTT DoCoMo and other MNOs gathered to create the O-RAN Alliance that we explained in the previous section. Figure 7 summarizes the new additional functions and interfaces that the O-RAN Alliance provides to supplement the 3GPP specifications.

We will explain the functions and interfaces in Figure 7 in the next subsection. As an introduction to the following section, and to better understand the difference between 3GPP and O-RAN specifications, Figure 8 shows how the functions and interfaces are positioned in the RAN architecture. We can see that the specifications that O-RAN Alliance provides are more detailed than the more generalized 3GPP specifications. This is important, particularly in the multivendor environment. For example, the X2 interface becomes essential for multi-vendor networks to function seamlessly [16]. In addition, we know from the previous section that the 5G RAN needs to support SA and NSA deployment. The current 5G deployments are still NSA and it will be challenging to move to the future 5G SA RAN deployment if the X2 interface is not open [16]. Open RAN’s multi-vendor RAN system dream also helps reduce costs for MNOs by introducing more software vendors and diversifying the supply chains. Despite that, Open RAN standards still need to comply with 3GPP specifications because 3GPP’s specification is the universal standard for the 5G RAN technology.

However, Open RAN’s principles are not only applicable to the 5G RAN technology. The purpose of Open RAN movement is to define and build RAN solutions based on general-purpose, vendor-neutral hardware and software-defined technology in 2G, 3G, 4G, and 5G [16]. Even though Open RAN can be used for other cellular generations, the current O-RAN Alliance’s standardization still focuses on 5G because it is the latest trend that the telecommunication industry and companies follow. We will introduce how Open RAN is deployed in 5G RAN in the next section about O-RAN Alliance architecture. How the interfaces and functional split defined by 3GPP for 5G NR are integrated into the O-RAN Alliance architecture will be discussed in the following section.

## 5. O-RAN Alliance Architecture

Besides the whole progress that the O-RAN Alliance has made, the O-RAN Alliance is also supported by OSC, so we can directly see how the standards are implemented into open-source software.

Generally, the O-RAN Alliance’s architecture for Open RAN consists of eight main parts: Service Management and Orchestration (SMO), Non-RT RIC, Near-RT RIC, O-RAN CU (O-CU), O-RAN DU (O-DU), O-RAN RU (O-RU), O-RAN eNB (O-eNB), and O-RAN Cloud (O-Cloud). Figure 9 shows the interconnections of these building blocks in O-RAN’s architecture provided by WG1.

The reference architecture shown in Figure 9 shows the concept to enable the next generation of RAN infrastructures [7]. This architecture is the foundation of making RAN open, virtualized, and AI-embedded, which is desired by many MNOs. The O-RAN’s architecture will be a standardized reference system, plus a complimentary reference system for other architectures made by other parties, such as 3GPP and other RAN-related organizations.

### 5.1. SMO

SMO is responsible for RAN domain management in the O-RAN architecture [58]. In the nutshell, SMO functions as the Management and Orchestration framework in O-RAN [5]. SMO provides RAN support in FCAPS; Non-RT RIC; and O-Cloud management, orchestration, and workflow management. The SMO framework has several components and interfaces such as the Non-RT RIC, A1, O1, O2, and open FH M-Plane. The relationships between these functions can be seen in Figure 10.

The O1 interface enables SMO to support the FCAPS of other O-RAN network functions. On the other hand, O2 interface is used by SMO for O-Cloud management, orchestration, and workflow management. Open FH M-Plane is used for FCAPS functionality from SMO specifically to O-RU. The Non-RT RIC function is to support non real time intelligent RAN optimization and perform intelligent RRM. The interface between Non-RT RIC and Near-RT RIC is called A1. The responsibility of the Non-RT RIC is to facilitate the operation of Near RT RIC functions within the RAN via A1 interface. The A1 interface can support three types of services: policy management, enrichment information, and ML model management.

As shown in Figure 10, the Non-RT RIC is further divided into two sub-functions. The first function is the Non-RT RIC Applications (rApps), which are applications to perform RAN optimization [6]. Second is internal SMO framework functionalities that terminate the A1 interface and expose services to the rApps through R1 interface. Non-RT RIC framework also hosts data management and exposure; and AI/ML workflow. There is also an internal messaging infrastructure for communication between Non-RT RIC with other SMO functions.

O-RAN’s operation and maintenance (OAM) is directly related to O-RAN’s O1 and O2 interface of SMO. O-RAN’s OAM architecture points out management services, managed functions, and managed elements supported in O-RAN. Figure 11 shows the logical architecture of OAM. From the figure, we can conclude that O2 is an interface to manage O-Cloud while O1 interface will manage other O-RAN building blocks from Near-RT RIC to o-eNB. Currently, OAM itself has two use cases: O-RAN service provisioning and O-RAN measurement data collection. Service provisioning focuses on supporting connectivity in/between physical network function (PNF) and VNF. MNOs can also operate and maintain the network through service provisioning. O-RAN measurement data collection focuses on collecting measurement data from the network so that the Non-RT RIC can do AI/ML training/inference/analyzing for optimization.

Moreover, there are actually several OAM models and deployment options, which are flat, hierarchical, and hybrid management models [59]. In the flat management model, all the O-RAN architecture entities are managed by the SMO through the O1 interface. The O-eNB, O-CU, O-DU, and O-RU are called managed functions (MFs) or managed elements (MEs). In the hierarchical management model, some MFs are allowed to manage lower-level MEs. An example would be the O-DU managing the O-RU. Lastly, the O-RU management responsibility could be shared between O-DU and SMO. This model is called the hybrid management model.

From the previous section, It is clear that WG2 and WG10 are responsible for the components within SMO. As of now, the development of SMO is led by Windriver. Ericsson focuses on the development of Non-RT RIC. In addition, the development of OAM is led by Highstreet Technologies. In the G release of OAM projects, the configuration of Network Functions (NF) using YANG models over NETCONF is supported through the O1 interface. As a whole, the O-RAN SMO standard is currently still under development.

### 5.2. Near-RT RIC

The Near-RT RIC is a logical function that allows near-real-time control and optimization over the RAN elements and resources through data collection and actions over its interface [58,60]. In short, Near-RT RIC will receive the policy from Non-RT RIC, follow the policy, and adjust the RAN parameters based on the given policy. To do this, Near-RT RIC has A1, O1, and E2 interfaces. Data collection and actions are performed by Near-RT RIC through the E2 interface connected to E2 nodes. The controlling actions over the E2 nodes are directed by the policies and enrichment data that the Near-RT RIC receives from the Non-RT RIC via A1 interface, as we have explained before. In addition, the O1 interface enables FCAPS from the SMO to Near-RT RIC. Near-RT RIC’s architecture can be seen in Figure 12.

An E2 node is a general term for a logical node terminating the E2 interface [58]. Specifically, the E2 node could be an O-CU, O-DU, or O-eNB. From the Near-RT RIC perspective, the E2 interface is a one-to-many connection [5]. Only CP protocols are performed in the E2 interface. The E2 functions include two category groups: Near-RT RIC Services and Near-RT RIC support functions. Near-RT RIC services contain report, insert, control, and policy. On the other hand, Near-RT RIC support functions cover E2 management and Near-RT RIC service update.

In Figure 12, we can see that the Near-RT RIC has several components. These components can be divided into two major groups: the RIC platform and the xApps. The RIC platform includes all the supporting termination and management components. The database stores and provides data from or to xApp applications and from E2 nodes. The xApps subscription management combines all subscription and data distribution operations. Conflicting interactions from different xApps are resolved by the conflict mitigation function. Security function protects the Near-RT RIC from hazards coming through the third-party xApps. Management services cover FCAPS and life cycle management for xApps. Messaging infrastructure provides a common message distribution system for different components of the Near-RT RIC.

Conversely, the xApps component is the main component of the Near-RT RIC. xApps is a set of applications where each xAPP provides specific RAN functions for the corresponding E2 node. Examples of these functions are RAN data analysis and RAN control [6]. xApps are considered as third-party applications and can be implemented by multiple microservices [5]. The xApps component is used to enhance the RRM capabilities of the O-RAN architecture. xApps can also provide basic info such as configuration data, metrics, and controls.

Currently, RIC platform development is led by Nokia. HCL leads the development of xApps. There are eight xApps that available in the project: Anomaly Detection xApp, Bouncer xApp, HelloWorld xApps, Load Prediction xApp, Measurement Campaign xApp, QoE Prediction xApp, Traffic Steering(TS) xApp and KPI Monitor xApp. In H release, RICAPP also have a new HW-Rust xApp to support RUST framework. In [61], it demonstrates Near-RT RIC via E2 interface in OSC F release to modify RAN parameters in the E2 node. It shows E2 capability to do report and indication service.

### 5.3. O-CU

The is a logical node hosting the functions of RRC, SDAP, and PDCP protocols of the BS [58,62]. In O-RAN architecture, O-CU can be further divided into O-CU-CP and O-CU-UP. O-CU-CP consists of the CP part of PDCP and RRC. Conversely, O-CU-UP covers the UP part of PDCP and SDAP. O-CU has a lot of interfaces which include the E1, E2, F1, NG, O1, X2, and Xn interfaces. The F1, NG, X2, and Xn interfaces can each be divided into the control and user interface. O-CU’s architecture and its interfaces can be seen in Figure 13.

E1, F1, NG, X2, and Xn are all interfaces defined by 3GPP. O-RAN Alliance reuses the E1 specification defined by 3GPP and adopts the E1 interface to bridge O-CU-CP and O-CU-UP. The F1 interfaces are used to connect O-CU to the O-DU. Inversely, the NG interfaces link O-CU to the 5GC. X2 interfaces connect the O-CU to other eNB or en-gNB in EN-DC configuration. Lastly, Xn interfaces link the O-CU to other gNB or ng-eNB. Both X2 and Xn interfaces are adopted from 3GPP with interoperability profile specifications added.

### 5.4. O-DU

RLC, MAC, and high physical layers are hosted in the O-DU logical node [58,62]. Data segmentation/integration, scheduling, multiplexing/demultiplexing, and other baseband processing are in O-DU. In general, O-DU is divided horizontally into two parts: O-DU high and O-DU low. O-DU high handles the layer 2 (L2) functional blocks, which are the RLC and MAC layer, while O-DU low covers the layer 1 (L1) functional block or the high physical. The functional application platform interface (FAPI) standard interface specified by SCF is used so that these two parts can communicate with each other. Other interfaces related to the O-DU are the E2 interface, F1 interface, O1 interface, and open FHI (O-FHI). O-DU’s architecture and interfaces to other components can be seen in Figure 14.

In the H release of OSC, the 5G O-DU High component is divided into eight different threads as shown in Figure 15, each identified by a different color. The first thread is responsible for managing the overall O-DU High functionality, including the O-DU utility and common functions. The second thread, known as DU_APP, handles the configuration handler, DU manager, UE manager, and ASN.1 codecs. The third thread is dedicated to managing the Evolved-GPRS Tunneling Protocol (EGTP). The fourth thread covers the Stream Control Transmission Protocol (SCTP). The fifth thread is focused on the O1 module. The sixth thread is responsible for the 5G NR RLC UL. Notably, there is a separate block for MAC_SCH, indicating its detachment from the 5G NR SCH (Scheduling). The seventh thread manages the 5G NR RLC DL and most 5G NR MAC functions, including the 5G NR scheduler (SCH) functions. However, the lower MAC handler operates within a different thread, specifically the eighth thread.

Since O-DU Low handles the high physical, it is constructed mostly by L1 processing blocks as shown in Figure 16. For the shared and control channel processing, we can see that the downlink (DL) and uplink (UL) are each handled by a separate block, which totals to 4: Physical DL Control Channel (PDCCH), Physical DL Shared Channel (PDSCH), Physical UL Control Channel (PUCCH), and Physical UL Shared Channel (PUSCH). Both Physical Broadcast Channel (PBCH) and Physical Random Access Channel (PRACH) have independent blocks. The task scheduling is divided into two blocks: one for UL and one for DL. Task scheduling will manage all queued tasks and begin the processing operations correspondingly. Then, we have the L2 FAPI processing block to handle the FAPI protocol L2 interface request/response. On the O-RU side, we have the FHI procedure block. We also have the forward error correction (FEC) acceleration processing to handle FEC operations such as passing FEC requests to hardware and invoking callback function on acceleration processing completion. Last, we have the timing events processing block to handle timing-related operations. Ref. [62] shows details of each processing blocks.

The O-RAN Alliance architecture adopts the Functional Split 7.2x for the lower layer split between the O-DU and the O-RU [58,63]. This split is carried out between the resource element mapping and FFT/iFFT stacks [64]. Figure 17 shows the location of split 7.2x and introduces the concept of precoding, using PDSCH as an illustrative example. Precoding is a physical layer technology that supports multi-layer transmission in multi-antenna wireless communications. In O-RAN Alliance split 7.2 specification WG4, it can be implemented in DU or RU. There are two variations of this Split 7.2x, which differ depending on whether “7.2a” precoding is performed in O-DU or “7.2b” precoding is performed in O-RU. This split is chosen by the O-RAN Alliance while considering the trade-offs in RU complexity and interface throughput. The details of the trade-offs are studied in [11,28,64,65].

Connecting this O-DU and O-RU functions split is the O-FHI. There are four planes inside the O-FHI, which are the Control, User, Synchronization, and the Management Plane. As we have known, the CP is a protocol for passing control signals, including scheduling and beamforming commands, configuration parameters, configuration requests, status, and responses. UP transfers the user data, such as the DL, UL, and PRACH In-phase/Quadrature (IQ) data. The Synchronization Plane (S-Plane) is used for achieving timing synchronization between multiple units of equipment. For S-Plane, there are several clock models and synchronization topologies that are extensively explained in [63]. Finally, the Management Plane (M-Plane) handles maintenance and monitoring signals. The data flows in the O-FHI are shown in Figure 18.

WG8 is in charge of developing the O-DU specifications. O-FHI specifications are developed by WG4.

O-DU high is developed by Radisys while O-DU low is developed by Intel. In OSC H release, O-DU High introduces multi-scheduler algorithm support. Other features added in this release also include Inter CU Handover, and E2 enhancements that support E2AP V30. Currently, integration testing with O-DU Low is ongoing while waiting for the availability of the RU simulator.

For the O-FHI, Intel has finished implementing the C, U, and S planes into the O-FHI library in the OSC repository. O-RAN adopters can directly use this open-source library in developing their O-RAN compliant DU and RU. However, the M-plane has not yet been developed. In fact, the M-Plane specifications are still under development.

### 5.5. O-RU

The O-RU hosts a low physical layer and radio frequency processing functions [58,63]. Each O-RU can have one or more panels. Each panel will consist of one or more TX-antenna-arrays/RX-antenna-arrays. A TX-antenna-arrays/RX-antenna-array is defined as a logical construct for data routing and is related to the physical antenna described in the O-RU construction. An array element will include one or more radiators, the specification of which is written in IEEE Std 145-1993. As we have explained, the O-FHI and the O1 interface connect O-RU to O-DU and SMO, respectively.

O-RU supports the beamforming functionality [63]. Beamforming allows a wireless signal to be directed specifically to a receiving device using multiple antennas. Requiring multiple antennas means that this beamforming process can be performed in MIMO. We will discuss the benefits of beamforming in Section 6. O-RAN Alliance architecture specifies that the beamforming commands are implemented between O-DU and O-RU [32,66]. O-RAN has included the digital, the analog, and the hybrid beamforming technologies in its specifications. O-RAN Alliance also proposed four methods for beamforming, which are predefined-beam, weight-based dynamic, attribute-based dynamic, and channel information-based beamforming method [63,66].

In predefined-beam beamforming, the O-RU has predefined a table of beam indexes containing beam weights. The O-DU will send the information on which index should the O-RU use. Unlike the former, weight-based dynamic beamforming allows the O-DU to generate the beam weights in real-time for the O-RU to use. Nonetheless, this method requires the O-DU to know the specific antenna characteristics of the O-RU. Sparing O-DU to know the details, attribute-based dynamic beamforming lets the O-DU instruct the O-RU to use a specific beam index associated with definite beam attributes. O-RU is in charge of generating the beamforming weights based on these attributes. Similar to the former, channel information-based beamforming also grants the O-RU to generate the beamforming weights. These weights will be created based on the channel information provided by the O-DU. For the mathematical description of these beamforming methods, please look into [63].

Since the RU is majorly hardware-based, O-RAN Alliance only focuses to develop the software standards to control this component. Vendors need to follow the specifications provided by O-RAN Alliance in making their RUs to be considered as O-RU.

### 5.6. O-eNB

The O-eNB is defined as an eNB or ng-eNB that supports E2 interface [58]. In the nutshell, O-eNB provides the O-DU and O-RU functions in an integrated node while still keeping the O-FHI between them. The O-eNB will have the E2, NG, O1, S1, X2, and Xn interfaces.

### 5.7. O-Cloud

The O-Cloud is a cloud computing platform that will host relevant O-RAN functions [58]. O-Cloud can consist of individual or collection of physical infrastructure nodes of the O-RAN architecture such as Near-RT RIC, O-CU, or O-DU. In order to work properly, O-Cloud should also host the supporting software components and the appropriate management and orchestration functions. Inside the O-Cloud, computing resources, such as general-purpose CPUs and shared task accelerators, are pooled and brokered by an abstraction layer [5].

The O2 interface will link O-Cloud to the SMO. The O-Cloud organizes the services relating to O2 interface into two groups: Infrastructure management services and Deployment management services. Infrastructure management services are responsible for the management and deployment of the cloud infrastructure. On the other hand, the deployment management services are responsible for life cycle management of the virtualized/containerized deployments. In OSC the development of O-Cloud is taking place within the in INF project.

### 5.8. Use Cases

In this subsection, we will introduce the O-RAN architecture’s use cases. The O-RAN use cases are generally divided into category I and category II. The reason for the division is the complexity and prioritization involved. Category I use cases address more general use case and category II addresses more specific use cases where these use cases require specific requirements [67]. These use cases have been explained deeply in one of the O-RAN Alliance’s white papers [68]. Several new use cases and updated deployment scenarios of O-RAN architecture are also mentioned in [42]. This subsection will slightly explain these use cases, including those new ones.

In category I, there are two key use cases, which are white box hardware design and AI-enabled RAN [68]. For the former, its use case is low-cost RAN white box hardware design. The white box was initially developed by the O-RAN Alliance’s WG7 as a method to reduce the cost of 5G deployment, and generally, the whole industry chain [39,68]. Releasing this white box can also reduce the difficulty of deploying the 5G tech, thus making it more attractive for small and medium enterprises in the telecommunication industry to modify the design [68]. The white-box hardware design is mostly focused in O-DU and O-RU, plus an additional FH gateway. The white box hardware design requires support of O-RAN’s O-FHI, O-DU open-source software, and facilitates NFV in the cloud.

For AI-enabled RAN, many use cases are described in category I [68]. These are TS, QoE optimization, QoS-based resource optimization, and massive MIMO optimization. TS means a process of mobile load balancing [68]. It is a widely used network solution to achieve optimal traffic distribution based on desired objectives. TS is becoming increasingly important nowadays, as the mobile network has grown rapidly in recent years. Optimizing connection issues such as inefficient, low quality, low QoE, and slow feedback response using a traditional optimization system is no longer the best approach. Because of that, the O-RAN architecture has developed the TS leveraging O1, A1, and E2 interfaces, thus making the network more flexible and agile and reducing network intervention.

Another category I use cases for AI-enabled RAN is QoE optimization. As we know, a 5G network can provide several applications with high bandwidth consumption and extreme sensitivity to latency, such as applications in Cloud Virtual Reality (VR) [68]. The current QoE framework can not support such applications because of dynamic traffic volume and fast fluctuating radio transmission capabilities. To reduce this problem, O-RAN Alliance deploys a new QoE framework, a vertical application made for specific QoE prediction and QoE-based proactive closed-loop network optimization. This closed-loop optimization is performed in real-time, involving the software-defined RIC. Before QoE is reduced, the radio resources can be allocated more to users and apps who need the most urgent radio resources. Because of its capability to adapt to users’ needs, the QoE is optimized, and the radio resources are utilized more efficiently.

QoS-based resource optimization is the next category I AI-enabled RAN use case. The network needs careful planning and configuration to support the diverse 5G services and applications. However, the traffic demand and radio environment are dynamic. Default configuration and planning might not be sufficient to provide highly demanding requirements in extreme situations. QoS-based resource optimization could be used in these situations to optimize RAN resource allocation between users when all user’s requirements could not be fulfilled. Analytics function in Non-RT RIC might conclude that it is better to prioritize a group of users through close examination of congestion situations.

The last category I use case for AI-enabled RAN is massive MIMO optimization. The MIMO framework has been considered one of the key technologies for 5G [68]. Aside from being a key technology, the massive MIMO has also become a reason why the 5G technology requires energy more than the 4G [36]. To overcome this problem, O-RAN architecture has tried to optimize the massive MIMO framework. The objective of this optimization is to improve cell-centric network QoS and user-centric network QoE in a multi-cell deployment area [68]. If needed, this optimization can provide a multi-vendor massive MIMO deployment area with multiple transmission points. However, this area can only be provided by MIMO when using specific MNOs. When applying this kind of optimization, O-RAN architecture has several advantages, including the possibility to apply and combine non-RT and near-RT analytics; apply ML; and make decisions for various tasks related to Non-RT and Near-RT RIC. Because of this optimization, the Non-RT and Near-RT RIC can oversee the current traffic, coverage, and interference situation in a whole cluster of cells. When doing this massive MIMO optimization, the O1, A1, and E2 interfaces will give necessary data, policy, and configuration exchanges between two components in the O-RAN architecture.

In category II, [68] mentioned several AI-enabled RAN use cases. The first use case for the interfaces is RAN slice service level agreement (SLA) assurance [68]. The network-level slicing occurred in 5G as a process that can provide E2E connectivity and data processing tailored to specific business requirements. These requirements include several network capabilities, such as support for high data rates and low latency. These capabilities are specified based on SLA, which was agreed upon between the MNO and the customer. Mechanisms to ensure slice SLAs and prevent any violations are becoming more popular. By combining the AI and ML models into SLA assurance mechanisms through Near-RT and Non-RT RIC, O-RAN Alliance can raise the possibility of SLA slice assurance being fulfilled. This importance of SLA assurance mechanism can lead to further research. Network slicing process still has so many opportunities to be explored by MNOs, thus making the network more efficient and changing the way how MNOs do their telecommunication business.

Another category II AI-enabled RAN use case is context-based dynamic handover management for vehicle-to-everything (V2X). Nowadays, the application of V2X concept in real life has increased rapidly. Because of this rapid increase, the V2X applications can be found easily, such as in self-driven vehicles, seat entertainment, and traffic assistance. These applications of V2X can increase traffic safety, reduce emissions, and save more time when riding [68]. The V2X works based on real-time information about the driver, the road, and the traffic conditions. Regarding these V2X applications, the O-RAN architecture can do several things, such as collecting and doing maintenance of the past traffic and radio conditions, deploying and evaluating AI and ML-based applications that can detect or predict abnormal users’ behavior, monitoring the traffic and the radio real-time condition, and deploying real-time xApps that predict and prevent anomalous situations in UEs.

The AI-enabled RAN can also be used in flight path-based dynamic UAV resource allocation use case. Like V2X, the application of UAV has been applied in many industries, such as agricultural plant protection, power inspection, police enforcement, geological exploration, and environmental monitoring [68]. The 5G technology can provide a high-speed network for several applications that need low-altitude UAVs. In O-RAN architecture, the Non-RT RIC can provide necessary information about the aerial vehicle and everything around that vehicle, including flight path information, climate information, flight limitation area, and space load information. To provide such information, the RIC can deploy Unmanned Traffic Management (UTM) for constructing and training AI and ML models that should be applied in RAN. This information can also be provided when Near-RT RIC performs radio resource allocation for on-demand coverage for a UAV, by considering the radio channel condition, flight path information, and other information.

Another category II AI-enabled RAN use case related to UAV is for UAV radio resource allocation. In UAV control scenario, the O-RAN architecture meets the need for wireless resource adjustment [68]. Rotor UAV usually flies at low altitude and low speed, while carrying mounted cameras and sensors. These UAVs have asymmetry requirements between UL and DL services. For UL, the 5G network has to support service to receive the 4K high-definition video UP data from the UAV. On the other hand, there is only a small amount of control data interaction requirement. This introduces a new requirement for the gNB. In addition, the existing network management platform could not optimize parameters for individual users. In O-RAN architecture, Near-RT RIC function module provides RRM functions and radio resource requirements for different terminals to O-CU and O-DU

Besides AI-enabled RAN use cases published in category II, there is also the virtual RAN use case. The use case for the virtual RAN network is RAN sharing. RAN sharing is an efficient and sustainable way to reduce the network deployment costs while increasing the capacity and the coverage of the network [68]. O-RAN architecture can accelerate the development of RAN sharing solutions because of its open and multivendor architecture. The O-RAN architecture also enables each operator to separately control their own VNFs in shared hardware. Remote O1, O2, and E2 interfaces can be introduced so that each operator could use their own SMO and Near RT RIC to independently monitor and remote control their O-DU in shared network resources. O-RAN architecture introduces freedom and ease in the RAN sharing process

In the previous paragraphs, we have explained several AI-enabled RAN use cases. O-RAN prioritized some of these use cases in [42]. The prioritized AI-enabled RAN use cases are TS; QoS and QoE optimization; RAN slicing and SLA assurance; and massive MIMO optimization. There are also new additional AI-enabled RAN use cases introduced, such as multi-vendor slices, O-RAN dynamic spectrum sharing, RAN slice resource allocation optimization, local indoor positioning in RAN, massive MIMO single user/multi user-MIMO grouping optimization, O-RAN signaling storm protection, congestion prediction and management, and industrial IoT optimization. Another new use case introduced is the O-DU pooling use case.

## 6. Challenges and Future Research Directions

This section summarizes the challenges that MNOs may face when deploying and developing the O-RAN architecture and the future research directions in response to the challenges. We will divide this section into two parts. First, we mention issues regarding the O-RAN architecture building blocks, covering the design and implementation challenges and future research directions for each of them. After that, we will mention issues about the Open RAN field as a whole.

### 6.1. O-RAN Alliance Architecture Issues

In this subsection, we will mention some issues regarding the O-RAN Alliance architecture and also its implementation in OSC. A common problem that occurs upon the entire O-RAN building blocks is that the standard specifications are not finished being written yet by the O-RAN Alliance. We have mentioned several nodes that still have not complete specifications in Section 4, which are SMO’s Non-RT RIC ML, O-FHI M-Plane, O1 interface, and O2 interface. OSC’s code development for the currently available specifications is also not finished yet, as mentioned in Section 4. Some of them have not or have just started like the O-Cloud. Some of them are postponed so that OSC can focus on developing other nodes, such as the O-CU and the O-DU MAC SCH. Others are waiting for the official specification release, such as the M-Plane.

SMO has an issue regarding copyright with 3GPP. Since some of the O-RAN specifications also reference 3GPP specifications, the occurrence of this issue is inevitable. Currently, this copyright issue focuses on the OAM part. However, O-RAN Alliance WG1 also plans to resolve this issue on the overall O-RAN architecture level. Besides these implementation problems, SMO also faces some design challenges and future research directions that we will explain in the following paragraphs.

Network densification causes the management and optimization of neighboring cell relationships more complex. O-RAN-based proactive automatic neighbor relations (ANR) optimization is a well-known application of SON that is used to manage neighbor cell relationships, or called neighbor cell relation table (NCRT) [69]. An ANR technique is proposed that identifies the trends before handover failure, marks the cells for handover prohibition, and identifies time-based trends to remove and add back the cells in the NCRT table by [69]. The ML-embedded ANR model is made as an rApp. The rApp has tasks to give several data updates to its neighbor cell. These data will also be used by rApp to improve default cell removal and prohibit any possible handover policies by generating a new policy. The rApp will send the new policy to xApp in Near-RT RIC. After that, the xApp executes the policy on Open RAN network nodes. This design will minimize the handover failure, reduce the operational cost, and thus increase the network’s performance and QoS.

One of the problems in network optimization/automation is that even though the objective is clear, there currently needs to be a practical way to select which intelligence model should be deployed, where to deploy it, and which RAN parameters to control. A rApp called OrchestRAN is proposed [70]. The OrchestRAN framework offers to solve this. MNOs can directly specify high-level control/inference objectives, and OrchestRAN will automatically compute the optimal set of algorithms and where to execute them. OrchestRAN prototype is tested on the Colosseum. Experiment results show that OrchestRAN can instantiate data-driven services on demand at different network nodes and time scales. OrchestRAN successfully achieved minimal control overhead and latency.

Another method proposed by researchers for SMO is to improve the RRM. The proposed intelligent RRM scheme is called cell splitting. The application of cell splitting is made by [71], where the cell splitting is involved in the Open RAN architecture to prevent cell congestion. Long short-term memory (LSTM) recurrent neural network (RNN) is utilized to learn, thus making several predictions about the traffic pattern that could arise from a real-world cellular network in a densely populated area. From these traffic patterns, LSTM will detect any cell that may be congested. LSTM model is trained at Non-RT RIC using long-term data gathered from the RAN. This cell-splitting architecture has several open problems that need to be solved in the future, such as the dynamic capability and flexibility of the model, the complex inference model, security risks, the requirement of extreme data rate, and the interoperability of multi-vendor elements.

Near-RT RIC’s current implementation of the TS xApp relies on the currently available AD and QP xApps. TS is critical in the RAN. Better ML models with higher accuracy should be applied in the AD and QP xApps. This will improve the overall TS xApp performance. Another problem arises from the design of the present O-RAN RIC. RIC cannot understand the connections between the same subscriber or other UE. Instead, RIC only owns high-level insights in a connection. This increases the risk of false-positive alarms/actions, especially without IMSI/IMEI correlation across connections. Some more suggestions are also raised by [72] to improve xApp and RIC platform. These suggestions include a smaller xApp package, support for time-series handling, support for the message queue, scaling out method, more AI technologies (online training, reinforcement learning (RL), and federated learning), and hardware acceleration.

O-RAN’s RIC consumes a non-negligible amount of resources. Additionally, O-RAN’s RIC has forward compatibility limitations because it is coupled to specific implementations, such as Redis or Prometheus databases. As a result, applications must poll these databases frequently to find new agents. The work in [73] introduces FlexRIC to answer these problems. FlexRIC is a software development kit developed by OAI to build service-oriented controllers. It is made up of a server library and an agent library. FlexRIC has two innovative service models with proof-of-concept prototypes for RAN slicing and traffic control. Results show that FlexRIC uses 83%.

There are several existing research relating to Near-RT RIC xApps. A method that enables fault-tolerant xApps in the RIC platform, known as RIC fault-tolerant (RFT), is proposed by [74]. The RFT is deployed so that the Near-RT RIC is more tolerant towards faults, thus preserving high scalability. Results show that the RFT can meet latency and throughput requirements as replicas increase [74]. Another xApp called NexRAN is proposed by [75]. NexRAN is an open-source xApp that can perform closed-loop controlled resource slicing in the O-RAN ecosystem. NexRAN is executed on O-RAN Alliance’s RIC to control the RAN slicing in srsRAN. NexRAN testing is performed in the POWDER research platform. Testing results show that NexRAN can successfully perform the RAN slicing use case. An RRM xApp based on RL is proposed by [76]. The xApp dynamically adapts the per-flow resource allocation, modulation, and coding scheme to meet the traffic flow KPI requirements based on the network status. The xApp testing is performed on OAI LTE RAN in a small laboratory setup. It is necessary to conduct future testing to determine the performance in large-scale, heterogeneous, and non-stationary scenarios.

O-DU or the DU generally is one of the most discussed topics by wireless mobile communications enthusiasts. In Section 5, we have mentioned that O-DU is responsible for L2 and some L1 processing. It is also noted that accelerators are included in the O-DU specifications. Back to Section 2, we also have pointed out that through NFV, the operator can use a virtual DU (vDU). However, there is currently a problem with the DU implementation. Since vDU is deployed in a COTS server, most vDUs rely on x86 architecture. This is because most COTS servers still use the x86 architecture. The problem is that x86 architecture cannot meet the O-DU requirements under special cases like meeting the numerology 4’s 1-microsecond latency requirement and handling heavy UE traffic load. In these cases, the x86-based processing can get extra support from accelerators.

Regarding accelerators, O-RAN Alliance has already defined some specifications for the use of accelerators in the new documentation. In [62], the terms lookaside and inline accelerators are explained. However, these specifications still lack the details needed for standardization. The method to access or utilize those accelerators, the accelerator’s API design and usage, and the definition of whether the accelerator is “dedicated or not” are not defined yet by O-RAN. Since accelerators are used in O-DU low, and Intel is the company responsible for developing OSC’s O-DU, the only accelerator that is compatible with the O-RAN code reference in the market right now is only the field-programmable gate array (FPGA) by Intel. Nevertheless, some vendors implement the DU independent of Intel, such as using Nvidia graphics processing unit (GPU) and Xilinx’s FPGA [77] as accelerators. Qualcomm will also introduce its accelerator card to bring more competition to the market.

Using accelerators makes vDUs more expensive than regular DUs [5]. Another solution to x86’s need for accelerators is changing to a reduced instruction set computer (RISC) or RISC-V-based processors like ARM or C5 that have more powerful computation power for the L1 and L2 processing that O-DU needs to handle. This solution will eliminate the need to use accelerators. Besides having better computation power, ARM-based processors are also more cost-efficient compared to traditional processors [78]. We will dig into details about this comparison in the next subsection.

Aside from the accelerator issue, interesting research studied the RU–DU resource assignment problem. The construction and maintenance cost of the O-DU is proportional to the amount of O-DU resources the cluster of connecting O-RUs uses over the operating time. MNOs are looking for intelligent assignment strategies for minimizing this resource, thus saving energy and providing free physical resources to support other services. The research carried out by [79] mentioned that efficient RU and DU resource management can be achieved by embedding a deep RL (DRL)-based self-play approach in RAN architecture, thus making the operation of RAN more cost-effective. Future research could be directed towards RU-DU resource assignment with multiple resources and large-scale RAN, leading to cost and latency problems.

O-FHI’s M-Plane is currently still being defined, just as we have mentioned before. Regarding this problem, a design of an M-plane for the next 5G O-FHI is proposed [80]. It is planned that the M-plane will support the initialization, configuration, and management of the O-RU. They also embedded several functions for the M-plane, including a “start up” installation, software configuration and performance, software fault, file management, and many more. The M-plane is designed based on O-RAN’s YANG model and includes O-RU controllers powered by a network configuration (NETCONF) protocol over Secure Shell (SSH).

As it happens, the O-RAN Alliance has released a few specification documents for the M-Plane, which are [81,82]. Security in the M-Plane has received attention. The prior specification uses SSH version 2 with a simple public key and password authentication. This approach has weak security, does not meet industry best practices, and violates USG guidance [44]. Transport layer security (TLS) with public key infrastructure (PKI) X.509 certification usage in the M-Plane was recommended for improved security [44,83]. Nonetheless, this new additional specification is optional for the MNOs to consider. For further details, please refer to [81].

While the O-RAN Alliance has already specified the security protocol for the M-Plane, no requirements are defined about the CUS plane. Adding a security protocol would compromise the latency budget and performance. The usage of MAC security (MACsec) for the CU-plane security of the O-FHI is proposed [84]. MACsec will be added with an ephemeral key exchange in the CP, while MACsec authentication only is proposed for the UP. MACsec is an optional security protocol in the eCPRI, so this proposal aligns well with the O-RAN Alliance specifications. MACsec is based on IEEE 802.1X and provides a simple and fast security solution compared to the internet protocol security (IPsec) standard in 3GPP, which is not dedicated to speed. Theoretically, the throughput for small packets using MACsec is almost double that of IPsec. Future research could be directed toward proving this through field tests. Another option to be considered is using WireGuard.

As mentioned, 3GPP defines a flexible, functional split for DU and RU. O-RAN Alliance chooses the split 7.2 option in its architecture. Research by [85] shows that flexible, functional splits have more benefits than fixed split options. The research focused on placing RAN slices in a multi-tier 5G Open RAN architecture. The problem was formulated mathematically, considering different functional split options for each network slice separately. Results showed that flexible, functional split outperforms fixed functional split in utilizing physical network resources and satisfying different network slice requirements. Another work also suggested a dynamic functional split to solve the timing requirement of connecting an O-DU or O-RU with a DU or RU that does not follow O-RAN delay requirements [7]. Further research should be undetaken to determine whether the O-RAN Alliance should upgrade its architecture specification to support a flexible, functional split.

O-RU has an issue related to the O-FHI. Various vendors have followed the O-RU specifications that O-RAN has supplied. Nevertheless, some of these claims are false. Despite claiming that their RU is O-RAN compliant, these vendors still design it with their proprietary standards. Some do not even use the O-FHI library that OSC has provided. It poses a huge problem because they will fail when the RUs want to be tested with the O-DU. As a solution, these vendors will modify existing or create new codes either on the DU or the RU side. While this method solves the problem, following the O-RU standard at the start of the RU designing process is more recommended.

One interesting research direction for the O-RU is related to the beamforming process. The beamforming process has become one of the solutions to reduce total power consumption in joint optimization schemes [86]. It was made to decrease the number of BBUs required to handle network traffic, maximize the total input from users by defining physical resource blocks, enhance the precision of radio connection, and increase throughput and number of parallel connections in a given cell area [66,86]. It can also minimize energy costs in mobile cloud and network considering QoS. Despite all these benefits of beamforming, this process still needs high cost and complex design when applied in a real environment. To overcome these problems, 3GPP and O-RAN Alliance have developed several functional split options. As explained in Section 4, O-RAN’s architecture uses the 7-2x split option, which can enhance flexibility, scalability, and interoperability within the network. In Section 4, we have also introduced several beamforming methods proposed by O-RAN Alliance.

The last method proposed by the O-RAN Alliance—the channel information-based beamforming method—is a digital beamforming process and can be considered the best method out of these four methods. This method offers steerable beams using antennas with dedicated radio signals and radio paths, enabling it to provide a high number of beams that can be transmitted dynamically. O-RU generates these beams based on the channel information from O-DU. However, implementing the pre-coding in O-RU can produce inter-RU interference. To overcome this problem, a zero-forcing algorithm (ZF) is proposed [66]. ZF is an algorithm that can be implanted in the beamforming process and solves the inter-RU interference problem. The ZF beamforming is a beamforming method that can mainly be used in massive MIMO. This method allows transmitting and receiving data to specific users while eliminating interference from these users. It is concluded that ZF can reduce the channel interference in its interface so that the high transmission bitrate can fulfill the low latency requirements by utilizing the advantage of low latency in the FHI. However, ZF implementation still needs several further research. Even though ZF can reduce channel interference and the low latency requirements can be fulfilled, it is still unexplained whether the channel status information (CSI) also has met the low latency requirements for URLLC.

O-Cloud might not be mentioned as frequently as other parts of the O-RAN architecture in research papers. However, there is research that proposed deployment strategies for the virtualized O-RAN units in the O-Cloud [87]. With embedded intelligence, O-RAN is considered a SON. O-RAN system has to maintain its performance and availability autonomously while maintaining the quality of service. Self-healing is a key feature in intelligently handling and managing failures and faults in the network. The proposed optimized deployment strategy aimed to minimize the network’s outage while complying with the performance and operational requirements. Binary integer programming was used to optimize the placement of the VNFs and their redundant ones. The model evaluation showed that the proposed strategy could maximize the network’s overall availability.

AIMLFW is a new project within OSC led by Samsung and focuses on implementing AI and machine learning (AI/ML) into the 5G system. The project was initiated to address the need for End-to-end AI/ML framework Management and Platform functions. Since it is the newest project within OSC, the current status is still in the early phase of defining an AI/ML workflow on OSC.

### 6.2. Open RAN Field Issues

This subsection will discuss broader issues related to the Open RAN field. Overall, there are three challenges that MNOs may face regarding the Open RAN field, which are implementation and standardization problems, improving performance and cost, and securing Open RAN. This subsection will be closed by other open issues that may probably occur in an Open RAN architecture.

Implementation and standardization problems are the first issues that we will discuss. Standardization is directly related to the O-RAN architecture because, as mentioned in previous sections, the O-RAN Alliance has made the most progress in developing the Open RAN framework compared to the other Open RAN projects. O-RAN Alliance has also provided detailed reference design specifications, complete with very clear documentation for each reference design. Even though O-RAN has provided such clear documentation, the alliance lacks robust, deployable, and well-documented software. Most of these O-RAN frameworks can not be used in real life, in actual networks [4]. It is caused by several reasons, such as the open source components proposed by O-RAN Alliance being incomplete, requiring additional integration, requiring further development for actual deployment, or containing components that lack robustness.

The difficulty of keeping pace with recent architecture standards is also a challenge. The cellular network community feels pressured because they have to keep updating themselves to the specifications and technologies that have just recently been developed in new communication, networking, and programming standards [4]. The most widely known examples are the NR and the mmWave architecture, which were introduced as 5G enablers by 3GPP. Both of these architectures are currently being deployed in closed-source commercial networks. The current problem with these developments is that the RAN software libraries (OAI-RAN and srsRAN) still need to be fully developed to support NR because these developments require further coding and compatibility testing. The same thing goes for mmWave, where testing the mmWave architecture for 5G technology is still impossible because of complexity. This complexity is caused by the lack of accessible open hardware for the mmWave software to run, and it needs several beam management tests for the software to run properly.

With the problems above, MNOs face dilemmas if they migrate their current RAN assets to Open RAN deployments. These decision points are summarized by [88]. The first question is whether MNOs should use the current Open RAN standard specifications, which still need to be completed, as we have mentioned, to open up the interfaces in vRAN or wait for the specifications to be more mature. The next question is whether MNOs should introduce Open RAN first in smaller networks before targeting macro RAN. While there is no simple answer and each operator will take different approaches, AT&T recommends that Open RAN implementation be introduced in incremental modules [44]. MNOs can launch Open RAN modules to their architecture bit by bit before changing the whole RAN architecture to Open RAN. Survey shows that small cells will be the initial focus of the vRAN deployment by MNOs [88]. Testing out in a localized network first before going to the main macro RAN would be a wise resolution [44,88].

The other question related to keeping pace with the Open RAN standard is whether MNOs should migrate their RAN assets to disaggregated and virtualized options simultaneously [88]. The counterargument is that it is less risky to successfully disaggregate the BBU first before considering cloud deployment. While the benefits of using both disaggregation and virtualization concurrently are attractive, the survey shows that 25% of MNOs will implement them separately at first [88].

Through openness, Open RAN has unlocked the possibility of interoperability amongst different vendors. Although economically beneficial, this opportunity poses new emerging challenges. Different vendors will use a wide range of components. It would be difficult to establish the location of bottlenecks that reduce the overall performance while using the heterogeneous components. In addition to the multiple vendor challenge, the requirement that Open RAN should also be able to interact with legacy 4G equipment will rocket the challenge even higher.

Some questions are also raised regarding the multivendor environment in Open RAN. When the RAN experiences a problem, how do MNOs decide which vendor should solve it? What is needed and how the operator decides in a multivendor environment remain open. There is also a concern about whether MNOs should directly deploy multivendor Open RAN or start working with a single vendor before reorganizing the supply chain to a multivendor environment at a future stage. Survey shows that about 40% MNOs expect to work with only one to-two established vendors at the beginning [88].

Integration challenge is the next issue relating to Open RAN implementation. Ericsson mentioned that it concerned Open RAN’s hardware integration [16]. With the complex multivendor ecosystem Open RAN offers, MNOs ranked integration issues as the second major obstacle with a 55% vote [89]. We know that openness has become a fundamental pillar of Open RAN from the very start [15], where in an OpenRAN architecture, hardware and software come from many different vendors, thus making it a multivendor architecture. Because the architecture involves components from different vendors, every service provider and mobile operator has to ensure that their networks continue to operate smoothly with all these different parties in their network [16].

There are four different Open RAN system integration models to tackle this challenge [16]. In the first model, MNO will do the integration, just like Rakuten and Vodafone did. However, this approach requires the MNO to possess great in-house expertise. Integration will be carried out by the service provider in the second model. This model would perform well if the service providers have extensive experience working with multiple resources. The third model assigns the integration role to the hardware or software vendor. Fujitsu shows an example of this, providing support in RU to DU integration for Dish [16]. Even so, it would be a challenge to ask vendors to integrate on behalf of their competitors [16].

The use of a service from a system integrator, a new actor, is the last model. The system integrator is a third party not associated with a certain hardware or software vendor, yet that works closely with the vendors to ensure their heterogeneous products work together [16]. Telefonica has provided an example by using services from Everis, a Spanish system integrator [16]. The appearance of this new role opens fresh business opportunities for companies who want to join the ecosystem. NEC, currently the most engaged system integrator, and others have sensibly taken this opportunity [90].

The RAN needs a new approach to guarantee that all MNOs can operate their networks smoothly in an Open RAN. This approach should prioritize a software-driven and open-minded ecosystem from hardware vendors, software vendors, system integrators, tower companies, real estate owners, regulators, industry bodies, and MNOs. The integration of Open RAN needs to be built for a software-centric world. In this software-centric world, each software talks to all physical components to deliver scalability and innovation and change how open networks are integrated. This approach will be mostly focused on Open RAN software, where the software will make those components smarter and interoperable. This approach will also help those components be integrated and maintained remotely by using a software upgrade, so MNOs and vendors no longer need to climb network towers. To integrate Open RAN, two levels of integration need to be carried out: Open RAN ecosystem integration and Open RAN software system integration. Ecosystem integration means the system integrator will be responsible for integrating the entire architecture, including open radios and BBU software. Meanwhile, software system integration is a process that mainly focuses on COTS hardware. In this integration, Open RAN can achieve its automation by using the same DevOps tools and the same CI/CD principles, thus simplifying the integration.

Integration and interoperability are critical challenges in Open RAN. Significant effort and collaboration are needed to test the Open RAN system. Meta mentioned that a standard development environment, standard optimization metrics, and standard test and validation methodologies are required to unlock the full potential of Open RAN [44]. It is where TIP, O-RAN Alliance, and other Open RAN standard organizations play an important role in helping mitigate this challenge. Testing methodologies, testing centers, WGs, and plugfests have been created, as discussed in Section 4. Future research should be undertaken in this direction. OSC offers standardization labs called OSC Community Labs [44]. Further information on the OSC Community Lab will be explained in Section 7.

A pipeline for designing, training, testing, and experimental evaluation of DRL-based control loops in Open RAN was proposed by [91]. The proposed pipeline is called ColO-RAN. ColO-RAN addressed the development and adoption of DRL in real networks. This problem was caused by the unavailability of large-scale datasets and experimental testing infrastructure. ColO-RAN enables ML research using O-RAN components. The capabilities of ColO-RAN were showcased on Colosseum and Arena testbeds. Performance evaluation of ColO-RAN was performed by developing three xApps to control the RAN slicing, scheduling, and online training of ML models. ColO-RAN and its related dataset will be publicly available to research communities worldwide.

The last issue relating to Open RAN implementation is to cross-examine Open RAN’s credibility in the promise of a multivendor environment and lower entry barriers for new vendors. Most leading Open RAN software companies have all based their products on FlexRAN, which is x86 dependent. Fortunately, the Open RAN ecosystem is expected to be more diverse. Nvidia’s BlueField-3 Arm-based data processing unit will compete against x86 processors. We also have mentioned several other competitors in the previous subsection. Amazon has already used ARM processors in some of its COTS, and other companies will presumably follow soon.

As with any other technology, the cellular network market and industry will try every possible solution that gives them better performance and lower prices. ARM claims that its server platform would provide up to 60% lower upfront infrastructure costs, up to 35% lower ongoing infrastructure costs, and up to 80% cloud infrastructure cost savings compared to traditional server [78]. Looking at the performance difference between ARM and traditional servers, inferring they are x86-based, ARM could become more popular. Furthermore, Arm has officially joined the 5G Open RAN policy coalition. Eventually, ARM vendors such as NXP and Qualcomm would emerge. Nevertheless, ARM products presently are more custom-built and are not general processors. In addition, Arm still lacks the flexible industrial-grade virtualization capability that rivals x86.

Improving the performance and cost is the common theme of the evolution process of the RAN, as we have explained in Section 1. This improvement process continues endlessly, including the current Open RAN era. One of the problems related to improving O-RAN and Open RAN performance is the resource scheduling process. The resource scheduling process is heavily related to network slicing. We have learned that network functionality is abstracted from its hardware and software components [4]. This abstraction leads to the formation of slices, and each slice is given a set of functions, services, and system resources, making it an independent, dynamically created logical entity that can support UEs with multiple needs [92]. However, each slice has been forced to carry out dynamic management due to increased demands. The management process has become critical and more challenging, especially regarding resource scheduling. Because of this reason, the slice needs new mechanisms that need to be developed, which requires a more optimized way of allocating resources.

There has been a lot of research that discusses improving resource scheduling and network slicing processes in Open RAN. Recent research by [92] provided a new algorithm in 5G’s slices called dynamic scaling multi-slice-in-slice-connected user equipment services for system resource optimized scheduling (DMUSO) algorithm. Like its name, the DMUSO algorithm uses a concept where, in an UE, slices are connected, so this concept is called multi-slice-in-slice connected UEs. This concept connects slices to services and resources to slices, not among slices in a UE. DMUSO algorithm can improve the 5G system performance by learning the service demands, data rates, resources, bandwidths, efficiency, transmission rates, and channel-related SLAs for a user equipment’s specific level in network slicing. Results showed that DMUSO achieves efficient and optimized system resource scheduling, with significant performance gains from 4.4 times to 7.7 times compared to other methods and algorithms. Further research related to DMUSO is still open, and [92] still has to analyze the effect of varying user equipment mobility on resource scheduling across slices in the future.

The potential significance of AI/ML in O-RAN 5G is believed to lie in its capacity to enhance system performance and improve the overall user experience. A Paper discussed the AI/ML workflow by providing details and illustrating the workflow of AI/ML in O-RAN-based 5G system architecture. The authors present a specific use case focusing on cell load prediction by creating a model through data training using the LSTM algorithm and integrated into xApps on the on the Near-RT RIC. The study shows enhanced performance in predicting the periodic traffic variations [93]. However, the workflow implementation remiains incomplete, with training data still requiring offline processing. A comprehensive implementation of the entire workflow is essential to demonstrate the effectiveness of AI/ML methodologies in closed-loop control for optimizing network performance within the O-RAN 5G framework.

There is also a resource allocation improvement issue in Open RAN. A resource allocation optimization model that can minimize the cost of updating a RAN infrastructure is proposed by [13]. The model allows the architecture to have a hybrid combination of different hardware and software generations by considering costs involving links, cell sites, and the capacity limit of RAN resources. Another method proposed to solve this resource allocation problem is hierarchical orchestration. Hierarchical orchestration is proposed to address over-simplified resource allocation and limited support for different network segments for the current one-size-fits-all orchestrators in E2E networks [94]. The E2E network is divided into three segments: the RAN segment, the transport network segment, and the core network segment. Every segment has its own distributed orchestrator. These distributed specialized orchestrators enable independent management of each segment.

Hierarchical orchestration also introduces a hyperstrator, a higher-level orchestrator to coordinate the distributed orchestrators and deploy the slicing process across multiple network segments. The hyperstrator is a central point, interacting with the whole E2E network. Therefore, hyperstrator has many tasks related to its orchestrators; one of them is to ensure cohesive performance across segments and slices to guarantee consistent QoS of the network slicing. From this proposed architecture, experiment results show that the hierarchical orchestration approach can leverage the capabilities of existing orchestrators and their communities to achieve a remarkable resource allocation in E2E networks. However, this research still needs further studies. Future research should be directed towards identifying requirements, classes, and relations of ontology for hierarchical orchestration protocols. Also, future research should define a systematic translation, make a model, and evaluate and assess the new hierarchical orchestration model.

A low-complexity, closed-loop control system for Open-RAN architectures to support drone-sourced video streaming applications is proposed by [95]. Flying drones has a higher likelihood of having line-of-sight propagation compared to UEs which can lead to performance degradation in high data rate transmission. The control system jointly optimizes the drone’s location in space and its transmission directionality to minimize its uplink interference impact on the network. The proposed system was prototyped and tested in a dedicated outdoor multi-cell RAN testbed. Numerical simulations are also used to evaluate the system. Results show that the control scheme achieves an average 19% network capacity gain compared to traditional BS-constrained control solutions. Building a 5G network that is energy efficient is also an important issue. Increased network capacity, geographical coverage, and increased traffic demands require network densification and will lead to more energy consumption. The negative impact of exhaustive energy consumption will not only degrade business profits but also impact the environment. State-of-the-art applications of ML techniques used in the 5G RAN to enable energy efficiency are reviewed by [96]. Recent research focuses on the functional split technique for green Open RAN [97]. An RL-based dynamic function splitting (RLDFS) technique that decides on the dynamic function splits among DUs and the CU in an Open RAN system to make the best use of renewable energy source (RES) supply and minimize operator costs was proposed. Performance evaluation was performed using a real data set of solar irradiation and traffic rate variations. Results showed that the proposed RLDFS method effectively uses renewable energy and is cost-efficient.

Other researchers also proposed F-RAN, or fog-computing-based RAN to improve RAN’s performance and cost. This is another type of RAN, which its architecture is based on fog computing. Fog computing is a term for an alternative to cloud computing that puts a substantial amount of storage, communication, control, configuration, measurement, and management at the edge of a network [98]. This fog computing can be applied to C-RAN to alleviate the constraints of FH and high computing capabilities in BBU pool [10]. F-RAN has several advantages, such as rapid and affordable scaling. This makes F-RAN architecture much more adaptive to the dynamic traffic and radio environment. Even though F-RAN is considered an excellent solution for covering C-RAN’s weaknesses, it still has some challenges and open problems

Another method researchers propose to improve RAN’s performance and cost is Air-Ground Integrated Mobile Edge Networks (AGMEN). AGMEN is proposed by [99] to integrate UAV-assisted network densification. In AGMEN, multiple drone cells are deployed flexibly to provide agile RAN coverage for the temporally and spatially changing users and data traffic. There are also several critical components in AGMEN, such as multi-access RAN with drone cells and UAV-assisted edge caching. However, AGMEN still has many challenges related to the heterogeneity and dynamic nature of architectural devices. Further research for AGMEN relates to mobile routing, multi-dimension channels, and UAV scheduling.

The use of Digital Twin(DT) has the potential to enhance the principles of intelligence, autonomy, and openness within an O-RAN-based system. As detailed in a study by [100], this application of DT technology is used to explore possible scenarios and train AI/ML models that can address all potential corner cases in a real-world setting. It will play a crucial role in achieving the predefined 6G KPIs.

The other method proposed by [101] to improve RAN’s performance and cost is PlaceRAN. PlaceRAN focuses on minimizing computing resources and maximizing the aggregation of radio functions to optimize the placement of radio functions in virtual NG-RAN planning. PlaceRAN uses a disaggregated RAN combination (DRC) concept and multi-stage problem formulation, thus enabling the management of units and protocols as a set of radio functions. The result achieved by [101] showed that PlaceRAN can reduce the number of DRCs in the network, by taking functional split options, one-way tolerated latency, cross-haul bandwidth, and a bandwidth derived from an analysis conducted in a project mentioned in the study.

Secure Open RAN is the next popular challenge that is widely discussed. Every network connectivity should be deployed as securely as possible, and so does the Open RAN and the 5G network. As users of network connectivity, we surely do not want to experience any fraud or data stealing that will threaten our safety. Therefore, a secure Open RAN should be seen as a big challenge, which should be discussed in further research.

Open RAN promotes the advantages of its disaggregation and interoperable pillars. On the other hand, these pillars also introduce new challenges regarding security. Disaggregation and virtualization are the themes of Open RAN. However, decoupling the software from the hardware expanded the security threat surface [102]. Major virtualization technologies are vulnerable to security attacks, including MEC, SDN, NFV, network slicing, and cloud [103]. O-RAN Alliance’s SFG has noticed that decoupling, containerization, and virtualization vulnerabilities can be exploited by threat actors [104]. Other works have also mentioned some of the vulnerabilities relating to disaggregation, such as insufficient identity, credential, and access management; insecure interfaces and APIs; system vulnerability; and shared technologies vulnerabilities [4,103]. The interoperable pillar of Open RAN also introduces new security challenges. One of them is that different vendors might use different degrees of security in their products [7]. Although vendors should practice best security practices, not all will implement adequate secure management interfaces.

The same case happens with the standards developed by the O-RAN Alliance. The O-RAN Alliance has two fundamental pillars: openness and intelligence. Each pillar introduces new security challenges. Additional interfaces and functions in O-RAN architecture add the area of threat surface [6,7,104]. Incorporating AI/ML in O-RAN architecture, known as the intelligence pillar, also adds a new surface threat targeting AI/ML-related functions in O-RAN. Other works have also discussed the new security challenges of using AI/ML in mobile networks [105,106]. Another additional threat surface comes from using open-source code. Open-source software gives both advantages and disadvantages regarding security [4,107]. However, extra security measures should still be taken because adversaries could easily access the same open-source code used in the O-RAN system and exploit its vulnerabilities [7,107].

As explained in Section 3, O-RAN Alliance is working carefully to address the security challenges with its SFG. SFG specifies the security requirements in the O-RAN system in [108,109]. Hopefully, the O-RAN Alliance can develop a secure architecture, framework, and guidelines for its Open RAN standards. Other references have also discussed some defense mechanisms to increase software security in addition to these requirements and protocols, such as authentication and access control, cryptography, secure virtualization, anonymity and obfuscation, resilience assurance, lightweight security based on the physical layer, and intrusion detection mechanisms [103,106,110,111]. SFG has also made modeling and analysis of security threats in the O-RAN architecture in [104]. SFG summarizes the threat surfaces, agents, potential vulnerabilities, and threats. Some parts of this analysis are also discussed by [112]. SFG specifies a total of 49 threats, divided into seven categories: threats to the O-RAN system, O-Cloud, open-source code, physical, 5G radio networks, and ML system threats.

Focusing on the O-RAN system, there are 35 threats [104]. We will mention some of the main threats for each O-RAN component. A common threat to all the O-RAN components is the exploitation of insecure design, weak authentication, and weak access control in O-RAN components or network boundaries to compromise O-RAN components’ integrity and availability. In the FH, the attacker could penetrate O-DU and beyond by accessing the O-FHI or targeting one or more planes in O-FHI. In O-RU, an attacker could set up a rogue O-RU. In Near-RT RIC, attackers could exploit xApps vulnerabilities or create malicious xApps. In Non-RT RIC, attackers could exploit rApps vulnerabilities or penetrate the Non-RT RIC to cause denial of service (DoS). Overload DoS attacks could also target SMOs.

O-RAN TIFG has also provided some E2E Security Test Specification in [113]. From an E2E perspective, the O-RAN system is just another gNB. Therefore, any O-RAN standard adopter must follow security requirements, threats, and test cases outlined in 3GPP TS 33.511 for gNB. Other than that, TIFG also specifies optional test cases for S-Plane, C-Plane, and Near-RT RIC A1 interface and O-Cloud. These test cases cover DoS, fuzzing, blind exploitation, and resource exhaustion attacks. Other references have identified some of these threats and attacks, such as data breaches, hijacking attacks, malicious insiders, data loss, DoS and distributed DoS (DDoS), abuse and nefarious use of services or resources, and caching attacks [103,105,110,114].

Another method proposed to improve the network security is deploying zero trust architecture (ZTA) [102,115,116]. Integrating ZTA to 5G and 6G technologies has become critical in tactical and commercial applications [116]. ZTA is a solution to address security requirements in a network with unreliable infrastructure. ZTA has a dynamic risk assessment and trust evaluation as its key elements. Intelligent ZTA (i-ZTA), an AI-embedded ZTA, is proposed by [116] to provide secured information in unreliable network infrastructure. ZTA can provide necessary computational resources and seamless, reliable, and robust connectivity.

Besides ZTA, another method proposed to deploy a secure Open RAN is called blockchain (BC). BC is a network system that can establish transactional faith among peer entities on decentralized peer-to-peer (P2P) platforms while overcoming the vulnerability of centralized ledger host [117]. BC has also emerged as a potential tool in designing a self-managed and scalable decentralized network [110,117]. The BC technology has been discussed by many researchers in Open RAN along with its various potential application scenarios, including IoT, MEC, smart grid, vehicular network, and smart cities. BC can offer pseudonymity, one of the defense mechanisms to keep its users’ privacy [118]. There are several main benefits if we integrated BC to Open RAN and its applications; one of them is reduced communication cost and reduced delay required to establish agreements facilitating a real dynamicity in RAN sharing [110,117,118]. However, incorporating BC into RAN still needs to be further studied. Future research should mainly focus on improving the latency, stability, and scalability of BC, developing a green mining mechanism for power-limited node devices, and finding ways to lower the cost of incorporating BC into RAN technology [110,117,118].

The following paragraphs will discuss other Open Issues relating to the Open RAN field. A lot of challenges and future research directions for Open RAN and O-RAN have been covered in the previous paragraphs. However, issues related to Open RAN still do not belong in previous categories. This last part will discuss those challenges and future directions.

As written in the previous implementation and standardization problems paragraphs, we can see that one challenge that needs to be researched deeply is the lack of accessible open hardware for software to run. The limited contributions to RAN open-source software intensify this issue. It seems like some more MNOs and vendors are trying to develop open-source CN and MANO frameworks, such as OMEC [119], Magma [120], Free5GC [121], and Open5GS [122]. At the same time, they are not trying to develop RAN-related project frameworks, like OAI-RAN or srsRAN. RAN development is mainly carried out by researchers and small companies with limited human resources and resources [4].

This limitation should be addressed because limited contributions result in limited feedback for the Open RAN standard developing organizations. It poses a severe problem since the organizations cannot verify the viability of the standard that has been developed. The major vendors and MNOs should be more involved in developing RAN-related projects because the RAN stack’s lower layers have become increasingly important each day. These lower layers have become sources of telecommunication businesses’ intellectual property and product-bearing revenues.

Related to this contribution problem is the different licensing models for 3GPP technologies. Different licensing models exist for 4G/5G technologies software, as shown in Figure 19. As mentioned, most RAN software, such as Amarisoft, Nokia, and Ericsson software, is closed-source, commercial, or copyrighted. Open-source software itself can be divided into permissive and copyleft licenses. Both allow developers to copy, modify, and redistribute code freely on open-source or commercial products. However, a copyleft license obligates the developer to open their altered source code under open source so it can be publicly available, such as the GNU General Public License (GPL), GNU Lesser General Public License (LGPL), GNU Affero General Public License (AGPL), and Mozilla Public License (MPL). This requirement is undesirable for companies using the source code because they must share their trade secrets. A permissive license does not require developers to release their source code using permissive licensed open-source software publicly. However, some permissive licenses do not retain the intellectual property rights. This creates potential future problems for companies using open-source software for their commercial products. Examples of this type of license are Berkeley Software Distribution (BSD), Massachusetts Institute of Technology (MIT), and Apache v1.1 License. Apache License v2.0 is an example of another type of permissive license that retains intellectual property rights.

These different licensing models show that companies can suffer a loss when using the wrong open-source software. Tackling this obstacle, OAI has its software license model to take OAI-RAN to be licensed under OAI’s public license, thus allowing their users to use their open-source software easily. The OAI public license allows parties to contribute to the source code while retaining their intellectual property rights. It effectively incentivizes the community to use OAI as a reference implementation in their research and development or productization.

Responding to this problem, O-RAN Alliance has also made its initial step to deploy, open, and softwarize RAN using Apache License v2.0. Through the initial step that O-RAN Alliance took, other wireless communities, MNOs, and vendors can follow O-RAN’s path by increasing their support toward developing complete and truly Open RAN and open-source RAN-related projects. Fortunately, more organizations are now expected to contribute to Open RAN. IEEE Standard Association has initiated an Open RAN Industry Connections Activity to help the existing Open RAN industry efforts [123]. This program will accelerate collaboration between organizations and individuals in the Open RAN field.

The next open issue related to RAN is the latency issue. Rakuten’s Open RAN deployment in Japan is a real-life example of this challenge. Even though the overall performance of Rakuten’s Open RAN mobile network is rated “very good”, the latency score shows room for improvement relative to major MNOs in assessed cities [21]. The latency problem is more severe in URLLC network slices that require low latency. Using vBBU could result in higher latency [71]. The following research should explore backup strategies for several use case scenarios.

A gradient-based scheme is proposed by [124] to overcome the latency problem. This scheme solves Open RAN’s minimum delay function placement and resource allocation. There are many layers of RAN, and Open RAN allows these layers to be split and deployed as virtual functions and openly communicate with each other layer for service provisioning. An E2E mobile operator employing Open RAN is modeled by [124], with a hierarchical mobile network architecture consisting of local, regional, and core data center layers. These three hierarchical layers add flexibility in resource allocation and increase reliability while taking advantage of Open RAN. In addition, the case where the traffic of a service function requests (SFRs) traverses multiple chains via a single path through containerized network functions (CNFs) is also modeled. This formulation proposed a gradient-based solution to achieve the optimal solution efficiently. This solution is applied through the gradient-based minimum delay (GBMD) algorithm. This algorithm can serve up to 90% E2E network delay decrease. Another method proposed by [125] can also be used to reduce latency for cross-service interactions in the network. The proposed method focuses on network slicing and is called optimized edge slice orchestration (MESON). This method is a combination of optimized cross-slice communication (CSC) in network-level slicing, MANO, and edge computing. MESON fosters cross-slice or tenant interactions, and provides the necessary means for the establishment of cross-service interactions, thus exploiting opportunities for providing co-location service. The research has shown that the main operations in the MESON layer can lower the delay of the service response time, with a delay of less than 40 ms, with points of presence (PoPs) of at least 100, thus reducing network latency.

Another issue that still exists in Open RAN is the issue of scalability. We already know that Open RAN is more scalable than its predecessors, thanks to NFV, SDN, and automation, which made the RAN more scalable and flexible in its management and orchestration [4]. Open RAN’s framework relies heavily on AI-embedded software to maintain its scalability and flexibility [126]. RIC is also responsible for making Open RAN scalable. Both Non-RT and Near-RT RIC have made the RAN more scalable and adaptable. The Near-RT RIC provides a secure and scalable platform, thus making it possible to control its xApps flexibly [39]. All of these are undertaken to make Open RAN more scalable, leading to a more effective RAN, especially in terms of its performance.

However, the scalability in Open RAN still needs to be improved. There are several concerns regarding the Open RAN scalability [22,89]:Operators wanting to adopt the cloud-native Open RAN need to expect the cloud scaling challenges [22]. Scaling the DU virtualization across servers will be a real-time problem. Additional scaling constraints will emerge if MNOs consider the use of accelerators. Furthermore, containerized orchestration alone cannot solve the high network availability and reliability operational goals. The applications will need additional built-in state synchronization and data integrity consideration. Moreover, specific failover and availability mechanisms will be required at the protocol level.Most Open RAN enthusiasts assume that the architecture will bring significant economic savings while ignoring the cost of operating the complex architecture [89]. We have explained that a variety of new business roles are involved in the Open RAN ecosystem, such as the system integrator. The question is whether the service expense for these roles and OpEx in a multivendor architecture at a large scale will exceed the cost-saving promised. The total TCO with large-scale operations is still undetermined [22].The gap between scalable automation and AI capability is another challenge to be considered [22]. The implementation of AI/ML technologies is relatively new in the telecommunication context. Using AI/ML in telecommunication grade operations will require significant resource expense. While interface standardization is currently available, data access, pipelines, and validation cannot be fully scaled due to a lack of standardized network configuration and performance data exchange. Furthermore, large-scale AI/ML asset deployment and operations by MNOs in live networks are still rare and complicated. To successfully scale and operate AI/ML use cases, harmonization of expertise from telecommunication, data science, and cloud/big-data fields is required [22].

OAI provides the Trirematics operator as a solution to answer the scalability challenge. Trirematics is an intelligent software operator for RAN and CN deployment scenarios [127]. Trirematics’ orchestration and management framework is cloud-native and extensible. Trirematics include intelligence, agility, automation, abstraction, maintenance, and observability. Trirematics makes intelligent and agile decisions in deploying and operating the E2E network. It automates the lifecycle of the network entities and abstracts complexities in a 5G environment while providing enough diagnosis and control for its users. It also has extensive observability features, such as log processing, alerting, metric processing, and health monitoring.

Rakuten Mobile’s commercial deployment in Japan proves that Open RAN is ready for prime time. Dell’Oro Group’s research report states that Open RAN is expected to acquire more than 10% of the RAN market share by 2025 [128]. While the majority respond enthusiastically, [5] view this result as proof to doubt the Open RAN adoption by the market. Whether these data and predictions are viewed positively or negatively is in the hands of each impacted party. Huawei clearly states that they do not support Open RAN or vRAN. Open RAN possesses many challenges, such as standardization, OpEx, and security issues that we have explained above. In addition, there might be political reasons involved behind the Open RAN movement. This notion was researched, and Open RAN was considered a geopolitical hijacking case [129]. By considering Open RAN as a social imaginary of openness and trustworthiness, Open RAN can be weaponized to outcompete rivals by parties ranging from industry to government. Whether this is true or false is still debatable; hopefully, the truth will shine as time progresses.

Whichever the case, AI/ML’s implementation will play an important role in the future of the cellular network. The 5G technology challenges show that more AI/ML is required in future network technology. O-RAN Alliance is applying this by building ML directly into the network architecture through RIC. We have also mentioned how researchers generally use ML to solve problems in O-RAN architecture and Open RAN. Several additional benefits of implementing AI/ML into 5G communication systems are mentioned in [130]. First, AI/ML can help with problems related to the complicated and dynamic wireless communications channel. AI can help in channel measurement data processing, channel modeling, and channel estimation. AI can also help in physical layer research of the 5G network. Massive MIMO arrays create enormous volumes of data well suited to be analyzed by ML systems. ML can also improve the performance in signal processing. Data-driven localization using ML algorithms in 5G systems can also be a valuable application. AI can also improve network management and optimization in 5G systems. Model-based optimization can be replaced with data-driven optimization in ultra-dense networks. AI/ML technologies also enable a paradigm shift of wireless networking from reactive to proactive design.

However, Ref. [130] also mentions some limitations of the current AI/ML technologies that prevent them from directly applying to the current 5G system. For example, ML algorithms are primarily developed for systems and applications that do not require high-frequency performance. On the other hand, a 5G network requires high data processing rates for URLLC. Existing ML algorithms also rely on powerful processing hardware. On the contrary, communication systems are full of devices constrained by storage capabilities, computational power, and energy resources. Adjustments and innovations in AI technology should be carried out before it can be fully compatible with the needs of the 5G network. One interesting research direction is to develop distributed ML algorithms for 5G networks. Previously, we have mentioned how MEC and fog computing could be solutions to improve the performance of Open RAN and 5G networks. The main idea is to move from centralized to distributed systems. Parallel to this trend, ML algorithms in communications applications should also move from centralized to distributed. For example, a lightweight deep learning model can support cloud, fog, and edge computing networks. Furthermore, parallel decentralized and centralized algorithms could be used while balancing complexity, latency, and reliability.

The 5G technology system is required to meet users’ demands worldwide by delivering connectivity anytime and anywhere [131]. Additionally, many smart devices and advanced technologies have been increasing rapidly, and these have also led to the increasing need for 5G connectivity systems. The 5G network is also demanded to fulfill multiple requirements. The development of optical wireless communications has been rapidly increasing because it could be a potential solution to support high data rates [132].

Moreover, 5G technology has also been required to provide services not only in big cities and metropolitan areas but also in remote and terrestrial areas—the areas that are uncovered or under-served geographically. A Non-Terrestrial Network (NTN) can be an effective solution for 5G to provide omnipresent services by achieving global network coverage [131]. NTN is networks, or segments of networks, that use an airborne or space-borne vehicle for transmission [133]. 3GPP has also carried out several studies and activities to support NTN as part of 5G technology, especially 5G NR architecture. NTN was explicitly introduced by 3GPP in Rel-17 [56]. The NTN in 5G NR is still developing, with a future guarantee from 3GPP that the 5G network can be operated in frequencies more than 52 GHz. We will know more about this guarantee in Rel-18.

The 3GPP standardization has already completed the first 5G NR standard, specifically in Rel-15 [56,131]. They have also started to work on several solutions to support NTN in 5G NR systems [131]. RRM has become one of the major issues in 5G NR, and this management strategy is relevant to offering tight cooperation between satellite and terrestrial networks. Through RRM, researchers are finding efficient resource allocation methods to decrease interference and provide real-time video services by implementing effective link adaptation procedures. Meanwhile, mobility management is also important to maintain the continuity of network services by achieving seamless handover over heterogeneous wireless access networks. The novelty of 5G architecture embedded with NTN is in the form of SDN and NFV-based-NTN; internet of space things (IoST), cognitive NTN; and non-orthogonal multiple access (NOMA)-based NTN [131].

## 7. Open Source Approaches

Open source RAN software enables the RAN software to be accessible to the majority of people, thus bringing innovation to the RAN technology. Currently, academics and research institutions can use open source RAN platforms, such as srsRAN, OAI, and OSC, to provide solutions to challenges faced in by the industry in RAN technology. As a result, more and more research is being carried out and more papers published that leverage the open source RAN platforms [33,46,47,74,75,91,134]. Some of these researchers also open up their work to be used for free, thus widening the open source RAN platforms available. These researchers help provide feedback on the shortcomings of open source RAN platforms and also on O-RAN Alliance specifications. The solutions proposed by these researches can also be used for new feature development in O-RAN specification. It can be expected that open source will be one of the research directions for the future of Open RAN.

Most 5G products currently use commercial software provided by software design house companies. Open-source RAN software is not used for commercial products. Open source can be used as a reference design or a proof of concept. However, with the rise of open-source RAN software platforms, we are starting to see the utilization of open-source software in commercial Open RAN products. BubbleRAN is an example of a company that uses open source as its core technology [135]. BubbleRAN is a startup company that aims to accelerate innovation in multi-vendor 5G through open source. BubbleRAN uses OAI’s experimental RT wireless platforms and Mosaic5G’s agile 5G platforms to develop their products. BubbleRAN’s products would offer the basic functionalities in the open-source platforms mentioned before. BubbleRAN provides extra features on their commercial products to support the features of particular solutions their customers require.

### 7.1. Interoperability and Security Testing

As we have mentioned in Section 6, interoperability testing and security testing of Open RAN are important issues to be addressed. Currently, the only available solution to ensure an interoperable and secure Open RAN system is the O-RAN Alliance’s standardized architecture description and testing procedure. There is no concrete testing system yet for vendors to ensure that their Open RAN products are interoperable and secure. Open-source testing software could solve the Open RAN interoperability and security testing problem. The cybersecurity domain has implemented open-source testing tool solutions, such as OSSEC, Kali Linux, and OpenVAS. We think the future direction of Open RAN security testing can also be directed into open-source solutions, following the example of the cybersecurity testing trend. These open-source security testing tools can provide the basic security testing requirements to meet the test case procedure requirements provided by O-RAN Alliance’s TIFG in [113]. Companies can still sell advanced testing services or products that provide extra features compared to their open-source counterparts.

The same concept could be applied for interoperability testing. Open-source interoperability testing tools can be developed to provide the minimum testing requirement for Open RAN products. Vendors can use this solution to ensure that their Open RAN products are interoperable with Open RAN other products. However, the system integrator role will remain the same due to the existence of open-source interoperability testing tools. The Open RAN multivendor ecosystem is still complex, whether open-source interoperability testing tools exist. Although MNOs can use open-source interoperability testing tools, using the services from system integrators might provide ease rather than forming an internal interoperability-focused team. System integrators can also provide advanced services for interoperability testing.

### 7.2. OSC Community Labs

OSC Community Labs or OSC Labs are integration and testing platforms offered by OSC. OSC labs provide the hardware and software resources for OSC members to test their software. OSC labs’ software follows the O-RAN specification. Currently, there are three OSC labs [44]. The first lab is located in New Jersey and is maintained by AT&T. It is the first laboratory in OSC and focuses on building an E2E environment using open-source software to demonstrate operational RAN. OSC New Jersey Lab has been accessible since early 2021 with the goal that is to creating a comprehensive environment where a functional Radio Access Network (RAN) can be showcased using open-source software. The second laboratory is located in San Jose and is hosted by China Mobile to maintain simplex and duplex instances of O-Cloud, allowing for seamless interconnection with the Bedminster Lab. This enables the sharing of testing tools and supports a continuous integration (CI)/continuous delivery (CD)/continuous testing (CT) pipeline built upon the XTesting framework. The third one is Taiwan Lab, which is maintained by the National Taiwan University of Science and Technology (NTUST) and National Yang Ming Chiao Tung (NYCU) and is supported by Chunghwa Telecom (CHT) and the ’B5G/6G Software Technology’ project funded by the Ministry of Science and Technology, Taiwan. The primary purpose is to replicate the environment of the Bedminster Lab, with a specific focus on integrating O-DU and conducting tests related to security [136]. Figure 20 shows the logical resources owned by each lab. OSC labs are important platforms for open-source RAN integration and testing. OSC labs can be leveraged for future research in improving open RAN.

## 8. Conclusions

Open RAN is accelerating innovation and disrupting the cellular industry ecosystem. It aims to overcome the challenges faced by the cellular network industry through disaggregation and interoperability. Open RAN promises to bring a multivendor ecosystem, flexibility, cost efficiency, and performance improvement by leveraging state-of-the-art technologies and approaches. Many projects, activities, and standardization efforts have been created for Open RAN. However, there are many challenges in developing and implementing the Open RAN idea into reality.

We have presented a comprehensive overview of Open RAN and its importance, discussed its challenges and potential research directions, and addressed some of the challenges from an open-source perspective. We first summarized the evolution history of the RAN from traditional to vRAN. It helps identify the differences and development of the RAN technology. Then, we introduced the Open RAN movement and the technologies related to the Open RAN generation. After that, we discussed projects, activities, and standards related to Open RAN. It helps to elaborate on the current condition of Open RAN development. Next, we briefly explained the standardized Open RAN architecture from the O-RAN Alliance and its use cases. After that, we discuss challenges and future research directions for each O-RAN’s standard architecture component. A number of broader issues relating to Open RAN and its future are also mentioned, including implementation and standardization, performance and cost improvement, security, and other open problems. Finally, we discuss how open source could potentially solve Open RAN challenges. Hopefully, this survey can be a useful starting reference for academia and industry to pursue further in-depth study on Open RAN.

## Figures and Tables

**Figure 1 sensors-24-01038-f001:**
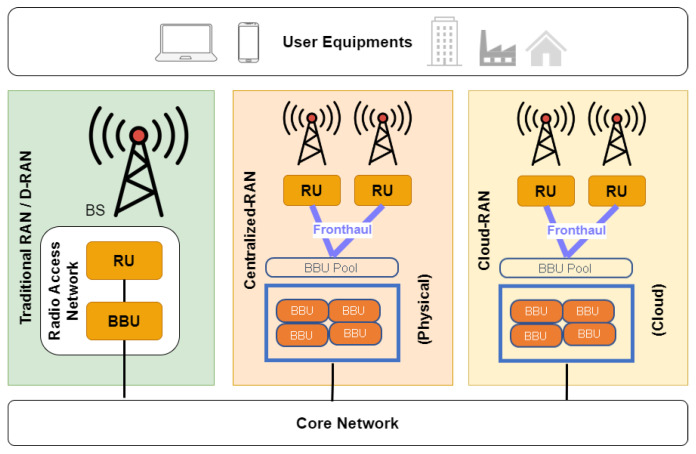
RAN architecture per generation.

**Figure 2 sensors-24-01038-f002:**
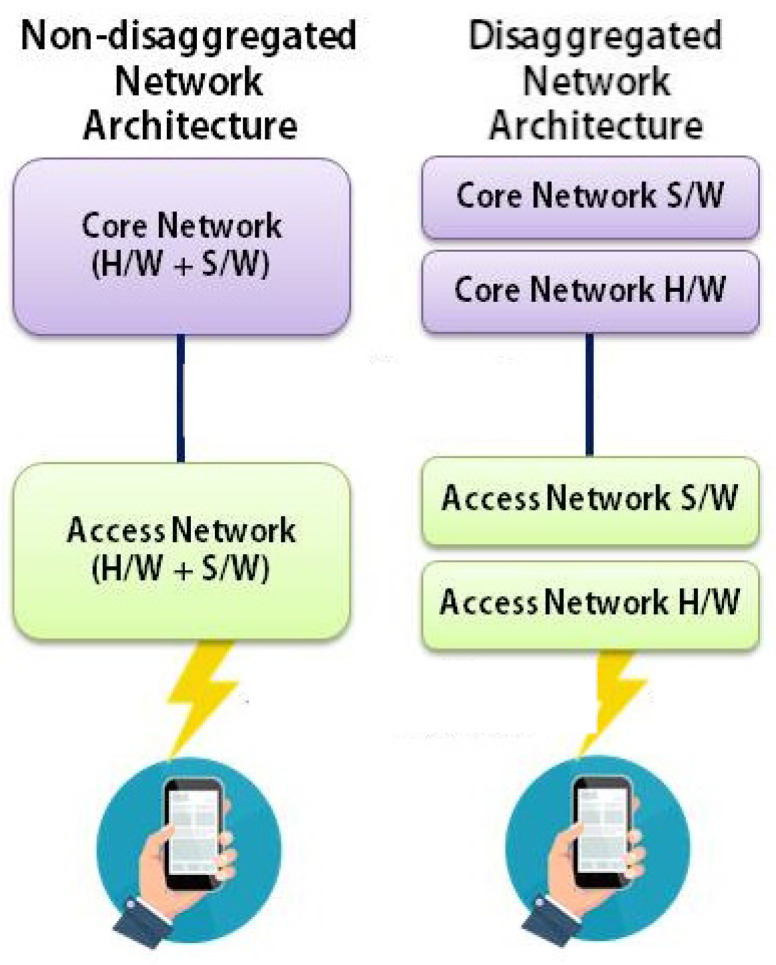
Comparison between a non-disaggregated and disaggregated network architecture.

**Figure 3 sensors-24-01038-f003:**
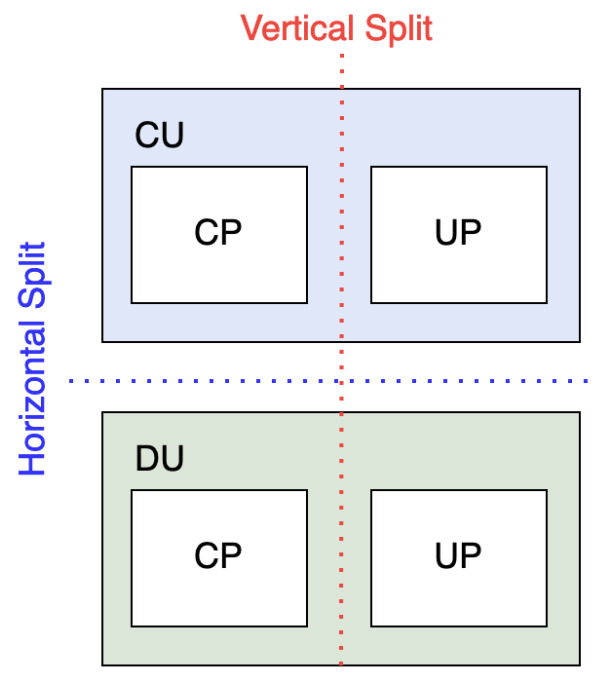
Vertical and horizontal functional split of Open RAN.

**Figure 4 sensors-24-01038-f004:**
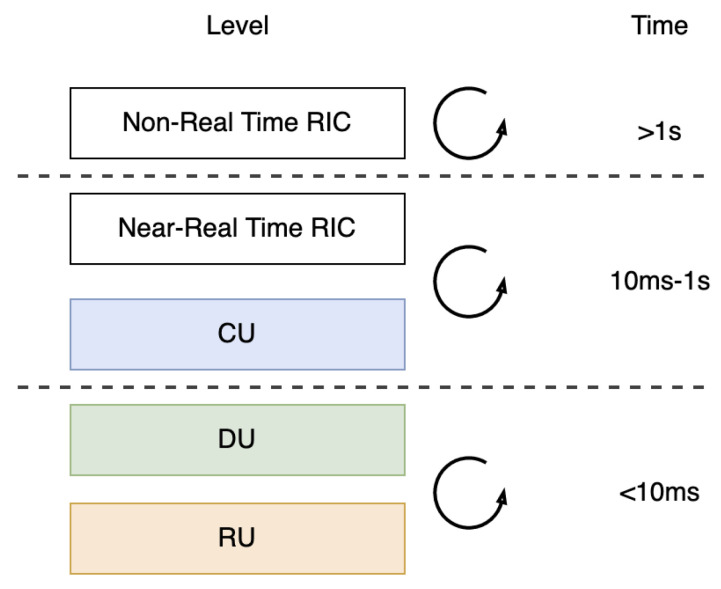
Three types of RIC.

**Figure 5 sensors-24-01038-f005:**
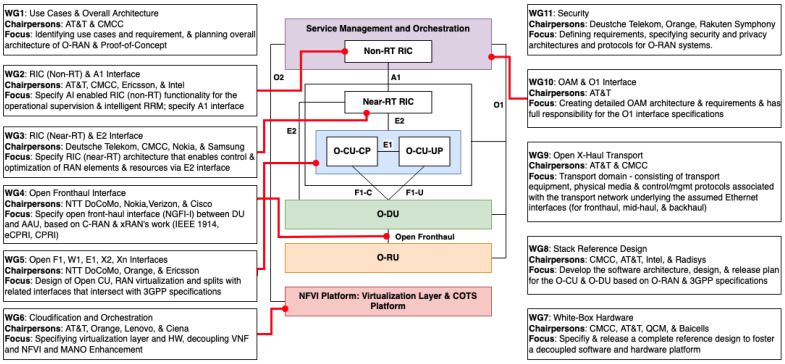
O-RAN Alliance’s WGs and their objectives.

**Figure 6 sensors-24-01038-f006:**
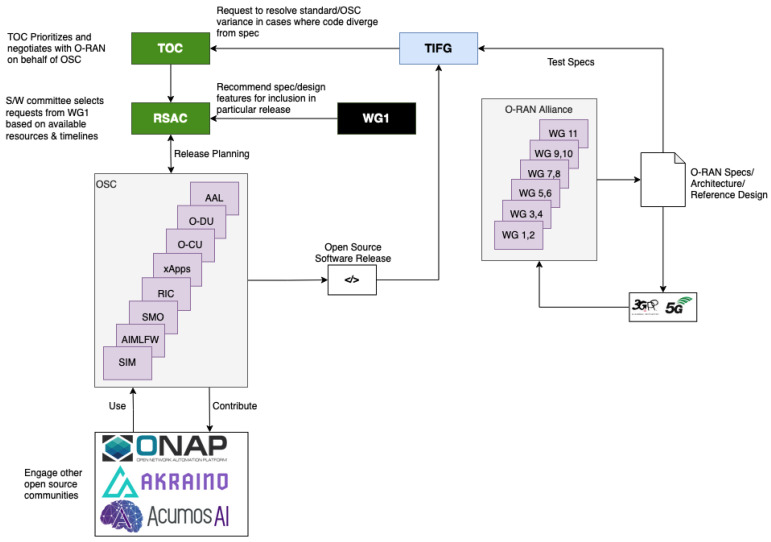
OSC’s workflow.

**Figure 7 sensors-24-01038-f007:**
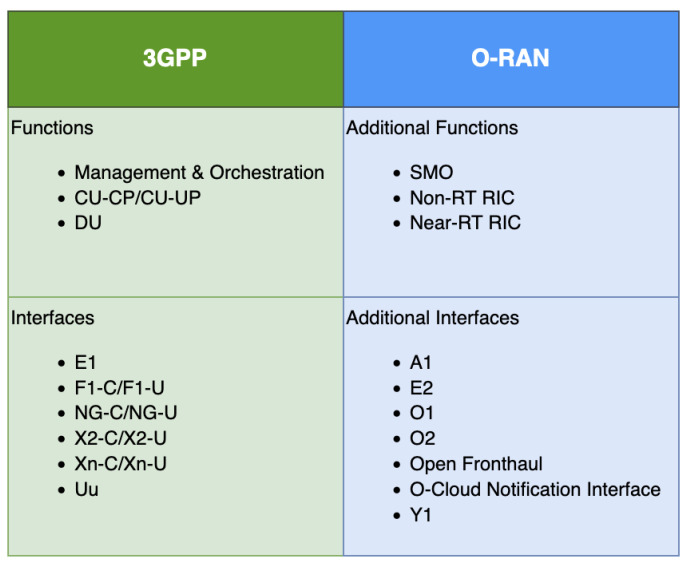
Comparison between 3GPP and O-RAN network functions and interfaces.

**Figure 8 sensors-24-01038-f008:**
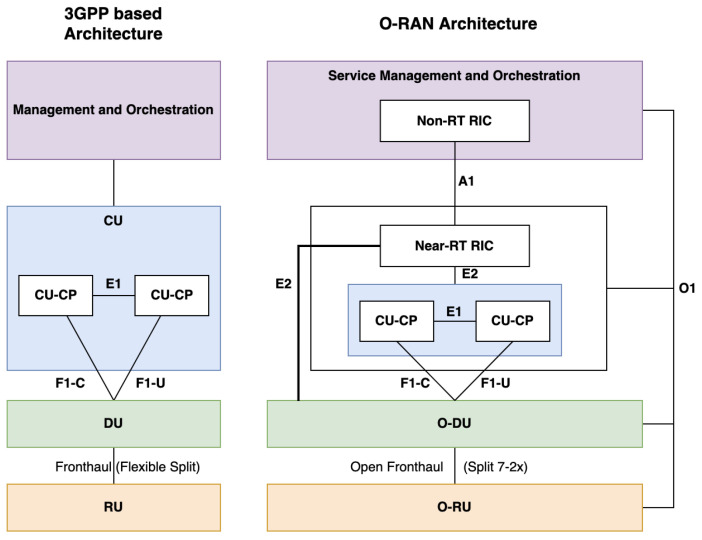
Comparison between 3GPP and O-RAN architecture.

**Figure 9 sensors-24-01038-f009:**
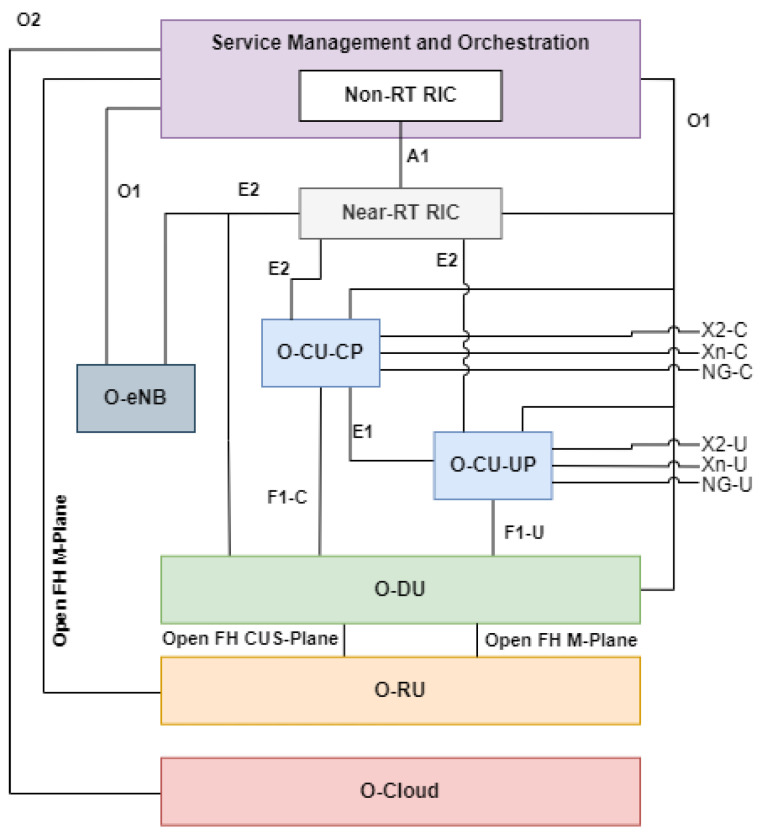
Logical architecture of O-RAN.

**Figure 10 sensors-24-01038-f010:**
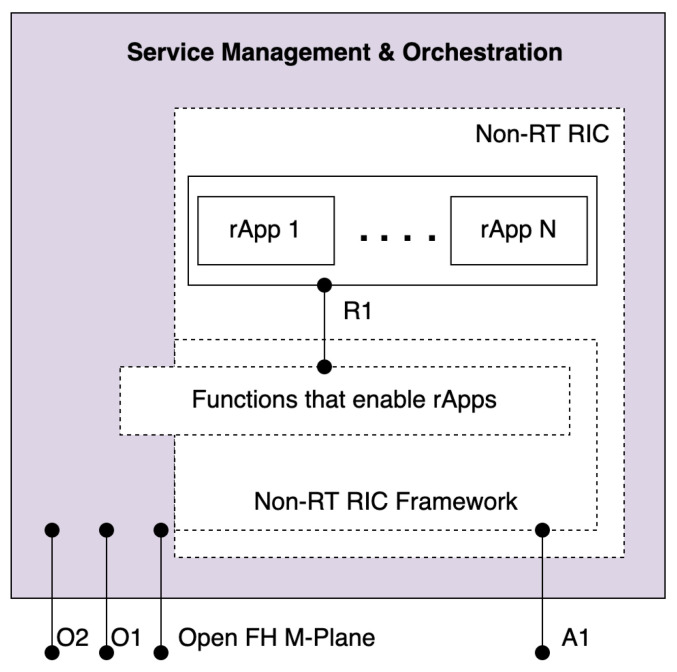
O-RAN’s SMO framework.

**Figure 11 sensors-24-01038-f011:**
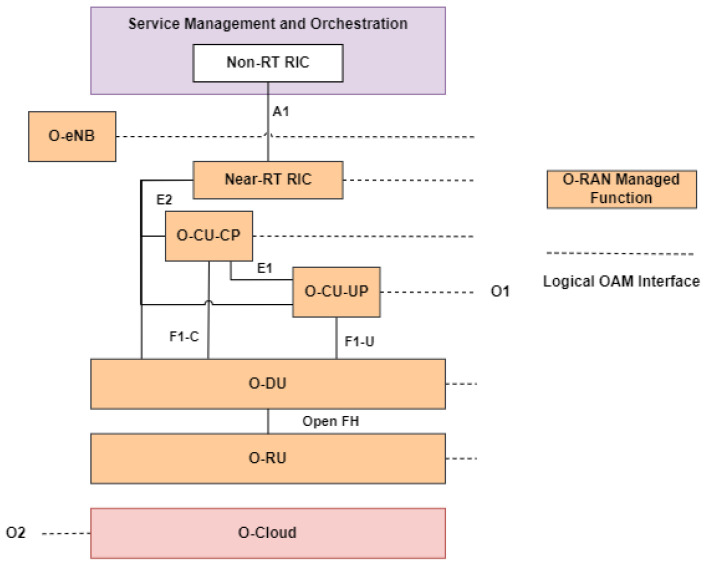
O-RAN’s OAM architecture.

**Figure 12 sensors-24-01038-f012:**
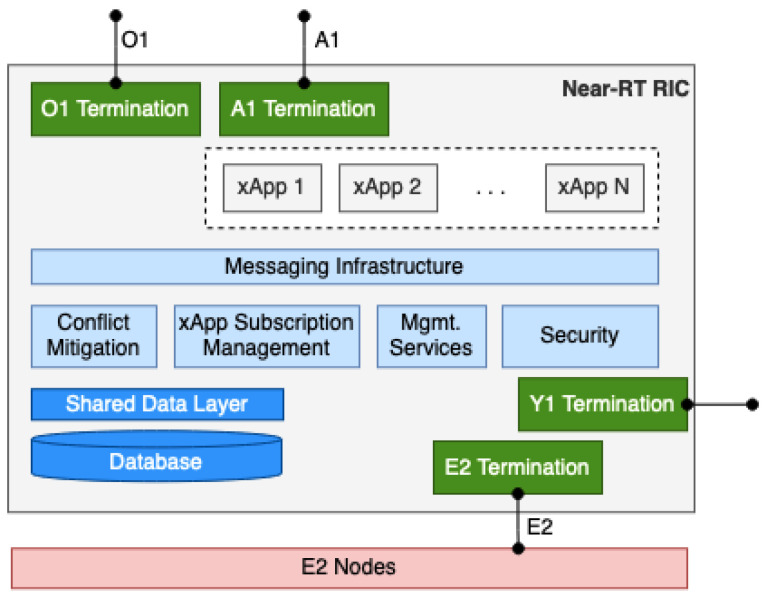
Near-RT RIC internal architecture.

**Figure 13 sensors-24-01038-f013:**
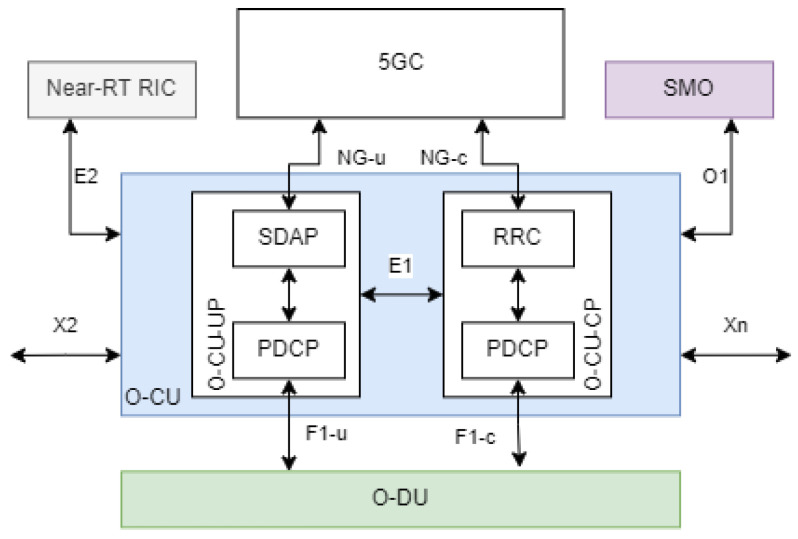
O-CU architecture.

**Figure 14 sensors-24-01038-f014:**
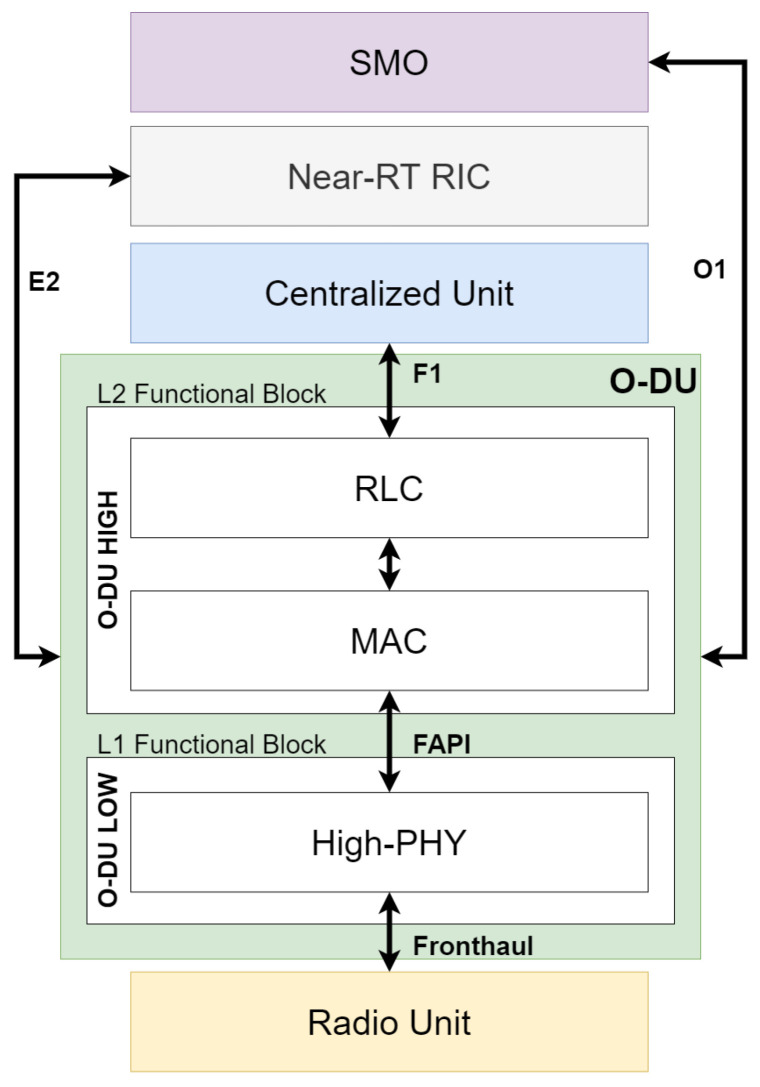
O-DU architecture.

**Figure 15 sensors-24-01038-f015:**
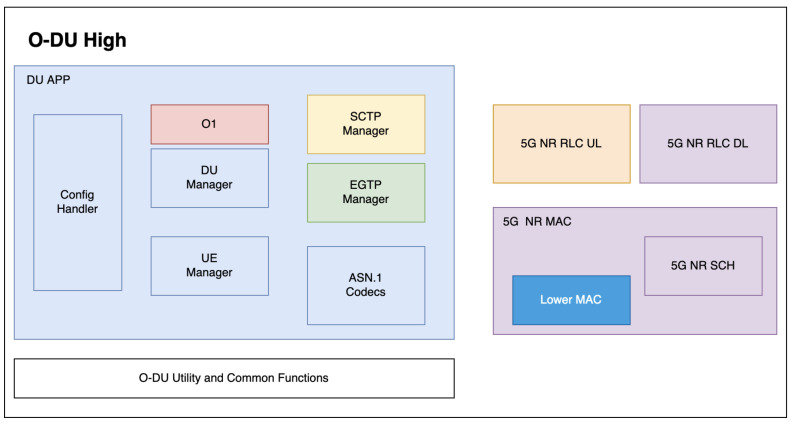
O-DU high architecture in H release.

**Figure 16 sensors-24-01038-f016:**
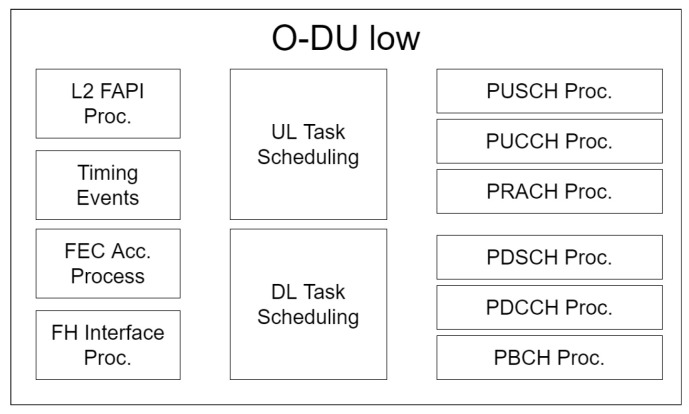
O-DU low processing blocks.

**Figure 17 sensors-24-01038-f017:**
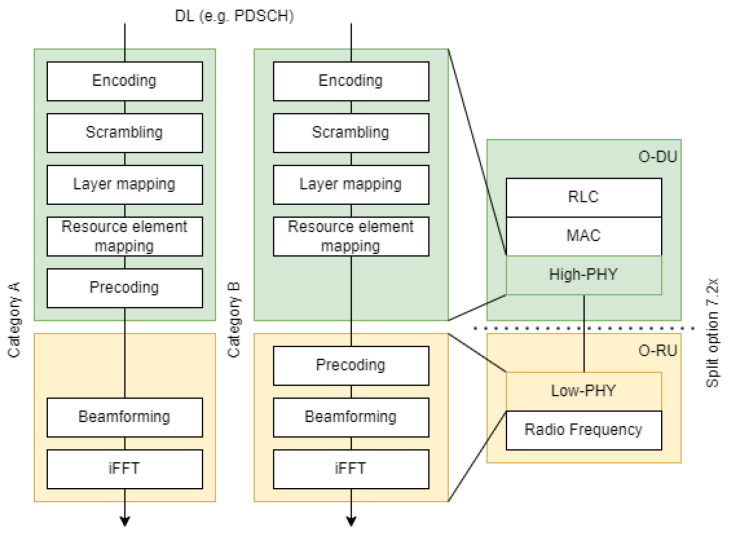
PDSCH in lower layer split 7.2x.

**Figure 18 sensors-24-01038-f018:**
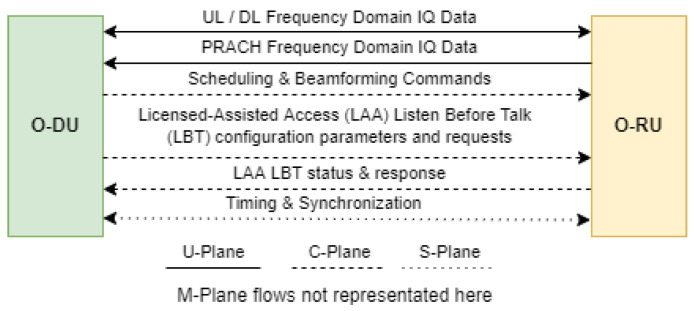
Fronthaul data flows.

**Figure 19 sensors-24-01038-f019:**
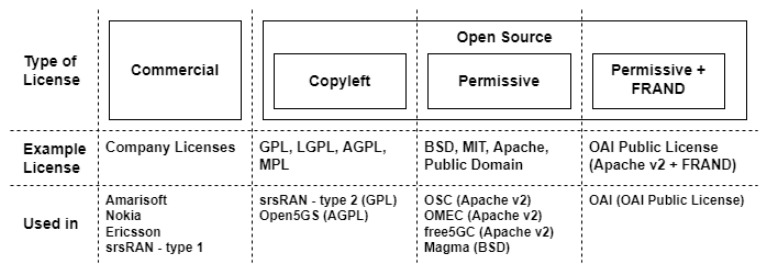
Open source 4G/5G software license types.

**Figure 20 sensors-24-01038-f020:**
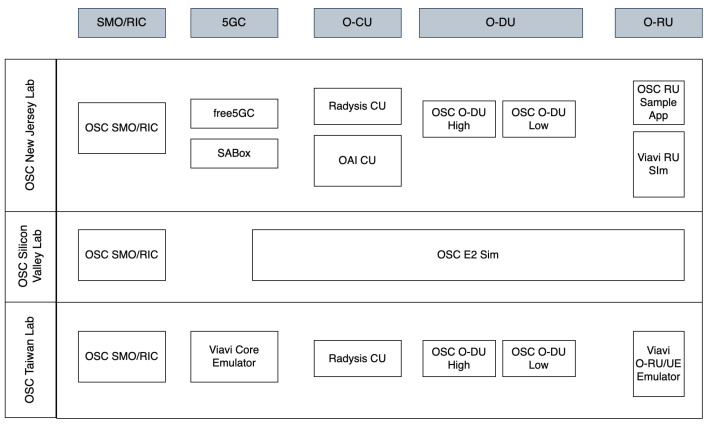
Logical resources of OSC Labs.

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
