# Peer review of "A Survey on Open Radio Access Networks: Challenges, Research Directions, and Open Source Approaches"

_sensors, 2024, doi:10.3390/s24031038_

Round 1
Reviewer 1 Report
Comments and Suggestions for Authors
The paper presents a somehow complete survey of O-RAN in a tutorial-based approach, managing to give the reader some updated knowledge about this new technology that is currently under development worldwide. After going carefully through the paper, this reviewer has the following comments, with the intention to be addressed as to enhance the technical quality of the paper.
Page 2, paragraph of lines 43-52: some phrases are repeated.
Page 5, lines 196-207: the authors's statements give the impression that O-RAN is already working up and running everywhere whereas, this is still a technology that is under development and standardisation, not yet a reality.
Page 8, line 321: the term peny-wise, possibly not appropriate for a technical paper.
Page 9, lines 378-379: This reviewer disagrees with the following statement: "However, it is predicted that AI in the future will still be an uncommon thing because the actual AI applications in real life is still rare".
Figure 9. For completeness with the text on the paragraphs, the figure needs the term "O-RAN cloud" to be positioned in the figure.
Make sure to use for all figures the same font and letter size. Also, take care of the readability of the text in the figures.
In Figure 17. Please express clearly which split option includes the precoding in the DU and which in the CU.
Page 26, line 968. Ml or ML models?, line 986: "police enforcement" is repeated.
Page 27, line 1019, TS stands for?, please add missing text.
Page 29, line 1126 microseconds instead of us.
There are several sentences written in present and future that refer to past dates, like if the text was outdated, for example: Page 33, line 1376: "It is impossible to avoid Intel’s products if an operator wants to build virtualized open RAN network in 2021". Page 40, line 1703: It is predicted that the whole vRAN market will see a 60% compound annual growth rate (CAGR) and the open RAN sector will experience 124% CAGR between 2020 to 2026 [88]. It is forecasted that the total cumulative open RAN revenues will hit 10 to 15 billion USD between 2020 to 2025 [126]. Please update the numbers and timeframes.
Page 36, line 1493: passion bandwidth?
The list of acronyms is incomplete, some missing: BC, RLDFS, RES, Col-RAN, etc. Please complete the list, probably in two columns to optimize space.
One important aspect that is missing in the survey is the use of the Digital Twin concept for O-RAN. Please add relevant information about this important aspect for the survey to be more complete and technically updated.
Comments on the Quality of English LanguageThe paper has uncountable grammar and typo mistakes. Here are only three examples:
Additionally, the challenge for RAN that still remains is the lack of efficient interconnection between each other parts of the network architecture [17].
This savings can be much higher than just 20%, because 5G network with these related technologies can give many business benefits that really significant for any business.
As written before, the third FG called OSFG is the parents of OSC. OSC is founded in April 2019 as a collaboration between O-RAN Alliance and LF [15], [40].
Author Response
Comment 1: Page 2, paragraph of lines 43-52: some phrases are repeated.
Reply to Comment 1: Thank you for Reviewer 1’s comments. We restructured the entire paragraph to make it more concise and remove the repeated phrase as you suggested
Comment 2: Page 5, lines 196-207: the authors’s statements give the impression that O-RAN is already working up and running everywhere whereas, this is still a technology that is under development and standardisation, not yet a reality.
Reply to Comment 2: We removed the section for the wrong impression statement and stick with the factual situation. Thank you for your kind reminder.
Comment 3: Page 8, line 321: the term peny-wise, possibly not appropriate for a technical paper.
Reply to Comment 3: We replaced the word with “economical” as we believed it is more suitable.
Comment 4: Page 9, lines 378-379: This reviewer disagrees with the following statement: “However, it is predicted that AI in the future will still be an uncommon thing because the actual AI applications in real life is still rare”.
Reply to Comment 4: After the authors discussed, we conclude that the context of “uncommon” is due to recent implementation on 5G. We rephrase the sentences according to your suggestion. Thank you
Comment 5: Figure 9. For completeness with the text on the paragraphs, the figure needs the term “O-RAN cloud” to be positioned in the figure.
Reply to Comment 5: The term O-Cloud is already positioned in the figure.
Comment 6: Make sure to use for all figures the same font and letter size. Also, take care of the readability of the text in the figures.
Reply to Comment 6: Thank you for your comments. We fix most of the figures to have better image quality and use the same font.
Comment 7: In Figure 17. Please express clearly which split option includes the precoding in the DU and which in the CU.
Reply to Comment 7: Based on Figure 17 we added the details of precoding is possible in the DU or RU. Thank you for the feedback.
Comment 8: Page 26, line 968. Ml or ML models?, line 986: “police enforcement” is repeated. Reply to Comment 8: We removed the repeated words and fixed the typo. Thank you
Comment 9: Page 27, line 1019, TS stands for?, please add missing text.
Reply to Comment 9: Thank you for your comments. The meaning of TS which is Traffic Steering already exists in line 759-765 thus we do not change sentence in line 1019
Comment 10: Page 29, line 1126 microseconds instead of us.
Reply to Comment 10: We replaced the word according to your suggestion. Thank you.
Comment 11: There are several sentences written in present and future that refer to past dates, like if the text was outdated, for example: Page 33, line 1376: “It is impossible to avoid Intel’s products if an operator wants to build virtualized open RAN network in 2021”. Page 40, line 1703: It is predicted that the whole vRAN market will see a 60% compound annual growth rate (CAGR) and the open RAN sector will experience 124% CAGR between 2020 to 2026 [88]. It is forecasted that the total cumulative open RAN revenues will hit 10 to 15 billion USD between 2020 to 2025 [126]. Please update the numbers and timeframes.
Reply to Comment 11: We decided to remove outdated statements accordingly. Thank you for the comments.
Comment 12: Page 36, line 1493: passion bandwidth?
Reply to Comment 12: PASSION is a project name. The bandwidth mentioned means a bandwidth derived from an analysis conducted in a project mentioned in the study. We rephrased the sentence accordingly. Thank you
Comment 13: The list of acronyms is incomplete, some missing: BC, RLDFS, RES, Col-RAN, etc. Please complete the list, probably in two columns to optimize space.
Reply to Comment 13: Thank you for Reviewer 1’s comments. We added BC, RLDFS, RES into the list. As for ColO-RAN it is a Developing Machine Learning-based xApps for Open RAN Closed-loop Control that we cited.
Comment 14: One important aspect that is missing in the survey is the use of the Digital Twin concept for O-RAN. Please add relevant information about this important aspect for the survey to be more complete and technically updated.
Reply to Comment 14: We added relevant information on the DT concept for O-RAN into the paper. Thank you for your suggestion.
Comment 15: The quality of “Figure 3. Vertical and horizontal functional split of open RAN” could be better.
Reply to Comment 15: Thank you for Reviewer 1’s comments. We replaced Figure 3 with the higher quality one.
Comment 16: [Grammatical Error] Additionally, the challenge for RAN that still remains is the lack of efficient interconnection between each other parts of the network architecture [17]. Reply to Comment 16: We rephrase and fix the grammatical error in this sentence for efficient interconnection among components within the network architecture. Thank you
Comment 17: [Grammatical Error] This savings can be much higher than just 20%, because 5G network with these related technologies can give many business benefits that really significant for any business.
Reply to Comment 17: We rephrase and fix the grammatical error in this sentence. Thank you
Comment 18: [Grammatical Error] As written before, the third FG called OSFG is the parents of OSC. OSC is founded in April 2019 as a collaboration between O-RAN Alliance and LF [15], [40].
Reply to Comment 18: We rephrase and fix the grammatical error in this sentence. Thank you.

Reviewer 2 Report
Comments and Suggestions for Authors
Authors have presented a comprehensive overview of open RAN and its importance, which can be published with some corrections. The volume of the submission is rather big, however, overviews are usually huge in volume.
According to the reviewer's opinion, the main drawback of the submitted overview is the absence of tables, which structure such huge overview material by categories and conclusions.
Some corrections:
The quality of "Figure 3. Vertical and horizontal functional split of open RAN" could be better.
In line 584 "scenario with with 256" one "with" should be omitted.
The quality of "Figure 10. O-RAN’s SMO framework" colud be better.
The meaning of the "more simplified communication interference" in line 1216 is imperceptible and need to be clarified.
In line 1223 "can be transmuted dynamically" should be corrected to "can be transmited dynamically".
In line 1234 "channel status information (CSI)" should be corrected to "channel state information (CSI)".
Comments on the Quality of English LanguageIn line 584 "scenario with with 256" one "with" should be omitted.
The meaning of the "more simplified communication interference" in line 1216 is imperceptible and need to be clarified.
In line 1223 "can be transmuted dynamically" should be corrected to "can be transmited dynamically".
In line 1234 "channel status information (CSI)" should be corrected to "channel state information (CSI)".
Author Response
Reply to Reviewer 2’s Comments
Comment 1: The quality of “Figure 3. Vertical and horizontal functional split of open RAN” could be better.
Reply to Comment 1: We replaced Figure 3 with a higher quality one. Thank you for your kind reminder.
Comment 2: In line 584 “scenario with with 256” one “with” should be omitted. Reply to Comment 2: We removed the repeated word. Thank you
Comment 3: The quality of “Figure 10. O-RAN’s SMO framework” colud be better.
Reply to Comment 3: We replaced Figure 10 with a higher quality one. Thank you for your kind reminder.
Comment 4: The meaning of the “more simplified communication interference” in line 1216 is imperceptible and need to be clarified.
Reply to Comment 4: The meaning of “more simplified communication interference” comes from the fact that, by separating the functions of phy layer and upper layers can enhance flexibility, scalability, and interoperability. We rephrase the sentence to make it clearer.
Comment 5:In line 1223 “can be transmuted dynamically” should be corrected to “can be transmited dynamically”.
Reply to Comment 5: We replaced the word “transmuted” with “transmitted”. Thank you
Comment 6: In line 1234 “channel status information (CSI)” should be corrected to “channel state information (CSI)”.
Reply to Comment 6: We replace the word according to your suggestion. Thank you
Comment 7: Please increase the quality of all your Figures (Figure 8 – interface is O1 I imagine).
Reply to Comment 7: We replaced Figure 8 with a higher quality one. Thank you
Comment 8: Moreover, additional works related to the AI/ML workflow, especially related to the intelligence loops that can be used for RRM, Orchestration, etc. are not highlighted in the survey
paper. In my understanding, the authors should include works related to AI/ML and specifically ML algorithms that are incorporated in O-RAN and utilized for optimization of RAN – beyond 5G/6G network parameters, typically based on Deep Reinforcement Learning methods, e.g.:
Reply to Comment 8: As you suggested, we highlight some works related to AI/ML into the paper. Thank you.
Comment 9: Please increase the quality of all your Figures (Figure 8 – interface is O1 I imagine).
Reply to Comment 9: We replace the word according to your suggestion. Thank you
Comment 10: In line 584 “scenario with with 256” one “with” should be omitted. Reply to Comment 10: We removed the repeated word. Thank you
Comment 11: The meaning of the “more simplified communication interference” in line 1216 is imperceptible and need to be clarified.
Reply to Comment 11: We removed the repeated word. Thank you
Comment 12:In line 1223 “can be transmuted dynamically” should be corrected to “can be transmited dynamically”.
Reply to Comment 12: We replaced the word “transmuted” with “transmitted”. Thank you
Comment 13: In line 1234 “channel status information (CSI)” should be corrected to “channel state information (CSI)”.
Reply to Comment 13: We replace the word according to your suggestion. Thank you
Comment 14: Although the manuscript is in general well-written, there are numerous syntax errors throughout the paper that need to be corrected for better comprehension.
Reply to Comment 14: We fix the syntax error on the section throughout the paper that we found. Thank you.

Reviewer 3 Report
Comments and Suggestions for Authors
In this manuscript, the authors present a thorough survey regarding current developments in the open radio access network (RAN) of next generation wireless networks. The authors first discuss the evolution of the open RAN concept (covering C-RAN, V-RAN, etc.) and then describe the technologies that are included in open RAN (e.g., network slicing, MEC, etc.), as well as present related projects and standardization activities. Finally, the authors illustrate the challenges and additional working directions that are required to complement the open RAN framework. The survey paper is in general well-written, interesting and relevant to the topics of Sensors. Some minor comments:
The authors should extensively proof-read their manuscript for typos and syntax errors. For instance, the sentences in lines 69-70, lines 133-134, line 152, lines 194-195, line 198, line 216 amongst others need to be reformulated.
Please increase the quality of all your Figures (Figure 8 – interface is O1 I imagine)
Moreover, additional works related to the AI/ML workflow, especially related to the intelligence loops that can be used for RRM, Orchestration, etc. are not highlighted in the survey paper. In my understanding, the authors should include works related to AI/ML and specifically ML algorithms that are incorporated in O-RAN and utilized for optimization of RAN – beyond 5G/6G network parameters, typically based on Deep Reinforcement Learning methods, e.g.:
1. Kaloxylos, A. Gavras, D. Camps Mur, M. Ghoraishi and H. Hrasnica, “AI and ML—Enablers for Beyond 5G Networks”, Zenodo, 2020
2. Xiong, Zehui, et al. "Deep reinforcement learning for mobile 5G and beyond: Fundamentals, applications, and challenges." IEEE Vehicular Technology Magazine 14.2 (2019): 44-52.
3. Karamplias, Theofanis, et al. "Towards Closed-Loop Automation in 5G Open RAN: Coupling an Open-Source Simulator with XApps." 2022 Joint European Conference on Networks and Communications & 6G Summit (EuCNC/6G Summit). IEEE, 2022.
4. Li, Jing, and Xing Zhang. "Deep reinforcement learning-based joint scheduling of eMBB and URLLC in 5G networks." IEEE Wireless Communications Letters 9.9 (2020): 1543-1546.
5. A. Giannopoulos et al., "Supporting Intelligence in Disaggregated Open Radio Access Networks: Architectural Principles, AI/ML Workflow, and Use Cases," in IEEE Access, vol. 10, pp. 39580-39595, 2022, doi: 10.1109/ACCESS.2022.3166160.
6. Y. Cao, S. -Y. Lien, Y. -C. Liang and K. -C. Chen, "Federated Deep Reinforcement Learning for User Access Control in Open Radio Access Networks," ICC 2021 - IEEE International Conference on Communications, 2021, pp. 1-6, doi: 10.1109/ICC42927.2021.9500603.
Comments on the Quality of English LanguageAlthough the manuscript is in general well-written, there are numerous syntax errors throughout the paper that need to be corrected for better comprehension.
Author Response
Reply to Reviewer 3’s Comments
Comment 1: The authors should extensively proof-read their manuscript for typos and syntax errors. For instance, the sentences in lines 69-70, lines 133-134, line 152, lines 194-195, line 198, line 216 amongst others need to be reformulated.
Reply to Comment 1: Thank you for the comments. We have reformulated sentences in lines 69-70, lines 133-134, line 152, lines 194-195, line 198 and line 216
Comment 2: Please increase the quality of all your Figures (Figure 8 – interface is O1 I imagine).
Reply to Comment 2: We replaced Figure 8 with a higher quality one. Thank you
Comment 3: Moreover, additional works related to the AI/ML workflow, especially related to the intelligence loops that can be used for RRM, Orchestration, etc. are not highlighted in the survey paper. In my understanding, the authors should include works related to AI/ML and specifically ML algorithms that are incorporated in O-RAN and utilized for optimization of RAN – beyond 5G/6G network parameters, typically based on Deep Reinforcement Learning methods, e.g Reply to Comment 3: As you suggested, we highlight some works related to AI/ML into the paper. Thank you.
